# Genome-wide analysis of DNA-PK-bound MRN cleavage products supports a sequential model of DSB repair pathway choice

Rajashree A. Deshpande[1], Alberto Marin-Gonzalez[2,3], Hannah K. Barnes [4], Phillip R. Woolley[4], Taekjip Ha [2,3] & Tanya T. Paull [4] ✉

The Mre11-Rad50-Nbs1 (MRN) complex recognizes and processes DNA double-strand breaks for homologous recombination by performing short-range removal of 5′ strands. Endonucleolytic processing by MRN requires a stably bound protein at the break site—a role we postulate is played by DNA-dependent protein kinase (DNA-PK) in mammals. Here we interrogate sites of MRN-dependent processing by identifying sites of CtIP association and by sequencing DNA-PK-bound DNA fragments that are products of MRN cleavage. These intermediates are generated most efficiently when DNA-PK is catalytically blocked, yielding products within 200 bp of the break site, whereas DNA-PK products in the absence of kinase inhibition show greater dispersal. Use of light-activated Cas9 to induce breaks facilitates temporal resolution of DNA-PK and Mre11 binding, showing that both complexes bind to DNA ends before release of DNA-PK-bound products. These results support a sequential model of double-strand break repair involving collaborative interactions between homologous and non-homologous repair complexes.

Double-strand breaks (DSBs) in genomic DNA are sensed by several evolutionarily conserved protein complexes that protect and repair the lesions. The most abundant of these in mammalian cells is DNA-dependent protein kinase (DNA-PK) composed of the catalytic kinase subunit (DNA-PKcs) and the DNA end-binding heterodimer Ku. The Ku component of DNA-PK appears in all kingdoms of life while DNA-PKcs is primarily found in higher eukaryotes[1]. Together, these factors protect DNA ends from nonspecific nuclease degradation and facilitate the recruitment of other factors involved in non-homologous end joining (NHEJ)[2].

NHEJ is considered to be the major pathway for double-strand break repair in mammals, although replication-dependent repair pathways that rely on homologous recombination for the resolution of processing intermediates are also essential for cell viability in dividing cells[3,4]. Homology-driven pathways require that DSB ends first be resected to remove hundreds of nucleotides from the 5′ strand, in order to facilitate loading of the Rad51 recombinase on the 3′ single-strands. The Mre11-Rad50-Nbs1 (MRN) complex is a critical component of this pathway as it performs the initial short-range resection events and also facilitates loading and activation of other nucleases that perform long-range processing[4–8]. The MRN complex, together with phosphorylated CtIP protein as an activator, performs this initial processing in the form of endonucleolytic cuts on the 5′ strand[6,8]. We and others have previously shown that efficient endonuclease cutting by Mre11 requires the Nbs1 protein, ATP hydrolysis by Rad50, and phosphorylated CtIP[6–8]. In addition, end processing by MRN (as well as the yeast MRX complex) requires a physical block on a DNA end which guides the Mre11 incisions to sites adjacent to the block[8–11].

During meiosis, it is clear that this critical protein block is the Spo11 protein—a topoisomerase-related enzyme that creates covalent linkages with DNA during meiotic prophase which require MRN(X) for removal and processing for successful strand exchange[12]. Outside of

[1]Encoded Therapeutics, 341 Oyster Point Blvd, South San Francisco, CA 94080, USA. [2]Department of Biophysics and Biophysical Chemistry, Johns Hopkins University School of Medicine, Baltimore, MD, USA. [3]Howard Hughes Medical Institute, Baltimore, MD 21205, USA. [4]The Department of Molecular Biosciences, The University of Texas at Austin, Austin, TX 78712, USA. ✉e-mail: tpaull@utexas.edu

meiosis when Spo11 is not present, however, it is not clear what constitutes the block. In previous work we have shown in vitro with purified recombinant proteins that the DNA-PK complex can play the role of an MRN-activating protein block, stimulating Mre11 endonucleolytic processing on the 5′ and 3′ strands adjacent to the end[9]. This collaborative function of DNA-PK in DNA end processing is perhaps surprising since the long-held paradigm for DSB repair suggests that NHEJ and replication-dependent pathways act in opposition to each other[13,14]. The widely used term "pathway choice" also has generally been used to describe a competitive relationship between NHEJ and homologous recombination and has implied a competition between DNA-PK and MRN for DNA ends.

Here we investigate the initial processing of DSBs by the MRN complex by analyzing DNA-PK-bound cleaved DNA fragments in human cells genome-wide. We find that the production of these processing intermediates is dependent on Mre11 endonuclease activity and occurs at sites of induced DSBs as well as at spontaneous endogenous sites. The pattern of DNA-PK binding and release at sites adjacent to DSBs depends strongly on the kinase activity of DNA-PKcs and correlates very well with the efficiency of binding of homologous recombination factors, including CtIP which we analyze here at DSB sites. Lastly, we use a light-activated Cas9 system for synchronized DSB induction to show that MRN and DNA-PKcs occupy the same ends prior to generation of DNA-PKcs-bound products. These results provide evidence for a sequential (rather than competitive) model of DSB repair in mammalian cells.

## Results

From previous work, we know that MRN/CtIP generates nicks on the 5′ strand adjacent to a bound DNA-PK complex, and can also generate double-strand breaks through endonucleolytic cleavage of both strands (summarized in Fig. 1A)[9]. To characterize these products of MRN/CtIP-dependent end processing associated with DNA-PK from human cells we used the inducible DivA system where ER-fused AsiSI generates double-strand breaks throughout the genome within 2 to 4 h of 4-OHT addition[15]. We established a modified ChIP protocol to purify DNA-PK-bound DNA fragments that are released from cross-linked chromatin by performing a gentle lysis, followed by immunoprecipitation with an anti-DNA-PKcs phospho-S2056 antibody, size selection, and purification (GLASS-ChIP)[16]. In previous work, we confirmed that these DNA-PK-bound DNA fragments are produced by human cells after AsiSI induction with 4-OHT using quantitative PCR[9].

Here we employed the GLASS-ChIP protocol with human U2OS cells expressing AsiSI, exposed to 4-OHT and DNA-PK inhibitor NU-7441 (DNA-PKi) for 4 h. We used DNA-PKi because we have found that inhibition of DNA-PKcs increases the yield of MRN-cut DNA fragments by several-fold in reconstitution assays with purified proteins and in human cells[9]. The DNA fragments from cells with or without 4-OHT addition were sequenced and GLASS-ChIP signal from the top 300 most efficiently cut AsiSI sites are shown (Fig. 1B). The result shows essentially no signal in the absence of 4-OHT, while a sharp peak surrounding the AsiSI site is seen with 4-OHT in the presence of DNA-PKi. Examples of genome browser views of several individual AsiSI sites are shown in Fig. 1C. The width of the DNA-PKcs peak in the presence of DNA-PKi is approximately 300 to 400 bp on average, covering 150 to 200 bp on each side of the AsiSI site that divides the peak (Fig. 1D, inset). We also observed many genomic sites where DNA-PKcs associates in the absence of DNA damage and found that these sites are often coincident with RNA Polymerase II promoters (examples of 4-OHT-independent sites in Fig. S1).

We have previously shown that nicking at sites of DNA-PK bound to DNA ends is Mre11 nuclease-dependent in vitro[9]. We also found in that work that the removal of DNA-PK from the ends of lambda DNA in single-molecule DNA curtains assays did not occur when an Mre11 nuclease-deficient mutant enzyme was used. In human cells, however,

it is challenging to eliminate all functional Mre11 activity since even 5% of the endogenous protein can carry out nuclease activity similar to wild-type cells and complete removal of Mre11 is cell-lethal[17]. To reduce Mre11 nuclease activity in cells we utilized the endonuclease and exonuclease inhibitors described previously, PFM01 and PFM39, respectively[18]. Exposure of U2OS cells to these inhibitors and DNA-PKi during 4-OHT induction of AsiSI translocation showed that loss of Mre11 endonuclease activity, but not exonuclease activity, substantially reduces the yield of GLASS-ChIP product at AsiSI sites (Fig. 2A, B). We also observed a similar pattern at sites of DNA-PKcs release from non-AsiSI locations (Fig. 2C).

## DNA-PK-bound chromatin fragments extend far from the break site

In the absence of DNA-PK inhibitor we previously observed 4-OHT-dependent GLASS ChIP recovery by qPCR, although the levels were several-fold lower than what is observed with DNA-PKi present[9]. Here we sequenced GLASS-ChIP libraries made from cells in the absence of inhibitor and found substantial recovery of 4-OHT-induced fragments, although the fold increase over background levels in the absence of 4-OHT varies substantially depending on the site (Fig. 3A). Aggregated data from the top 300 AsiSI sites is shown in Fig. 3B, with data from individual genes in Fig. 3C. One obvious difference between inhibitor-free samples and DNA-PKi-treated samples is the fact that DNA-PKcs GLASS ChIP signal extends much farther from the AsiSI break site in the absence of DNA-PK inhibition. Without inhibitor, the recovered fragments cover at least 1 kb from the AsiSI site, in some cases up to several kb, while signal is limited to less than 200 bp in the presence of DNA-PKi. This is likely related to the immobility of DNA-PK in absence of its catalytic activity. Previous experiments in vitro have shown that DNA-PK is bound nearly irreversibly to DNA ends in the presence of catalytic inhibitors or when the active site of DNA-PKcs is mutated[19–22].

The vast majority of GLASS-ChIP peaks in the genome are not dependent on 4-OHT exposure (examples shown in Fig. S2). In general, these DNA-PK ChIP signals are significantly higher than AsiSI-associated peaks and are tightly correlated with RNA Polymerase II, both the total polymerase and phosphorylated S2-RNAPII ChIP (RNAP2S2), based on comparisons of previously published ChIP data in U2OS cells[23]. This observation agrees with many reports of DNA-PKcs association with sites of active transcription and even a requirement for DNA-PK in transcriptional regulation in some cases[24–27]. These sites are often not associated with DSBs, as shown by a lack of BLESS signal, and also no accumulation of other DSB markers such as Lig4, 53BP1, and Rad51[28] (Fig. S2A). In other cases, the DNA-PKcs GLASS-ChIP signal is found at sites that also show several other markers of DNA double-strand breaks, including Lig4, 53BP1, and BLESS signal (examples of both types of sites in Fig. S2B).

To compare the yield of DNA-PKcs-bound fragments released by end processing to the pattern of DNA-PKcs occupancy in the genome, we performed conventional DNA-PKcs ChIP using phospho-DNA-PKcs antibody with extensive sonication of chromatin (DNA-PKcs "Pellet-ChIP") in the absence of DNA-PKi. Recovery of DNA-PKcs-bound pellet ChIP fragments at the top 300 AsiSI sites appears to be qualitatively very similar to what is recovered from the released GLASS-ChIP fragments (Fig. 4A), and a comparison to the GLASS-ChIP recovery of fragments at these 300 sites shows a correlation coefficient of 0.79 (Fig. 4B). Genome browser views of the released versus chromatin-bound DNA-PKcs show that the patterns are generally very similar, with many sites independent of 4-OHT as well as AsiSI-associated peaks (+4-OHT, Fig. 4C). Higher resolution views show that, in some cases, the spreading of DNA-PKcs away from the AsiSI site is more pronounced with the GLASS-ChIP products (Fig. 4C panels 2 and 4). Consistent with these observations, the average width of the GLASS-ChIP peak is approximately 850 bp on either side of AsiSI, while the corresponding width in the pellet-ChIP samples is approximately 700 bp.

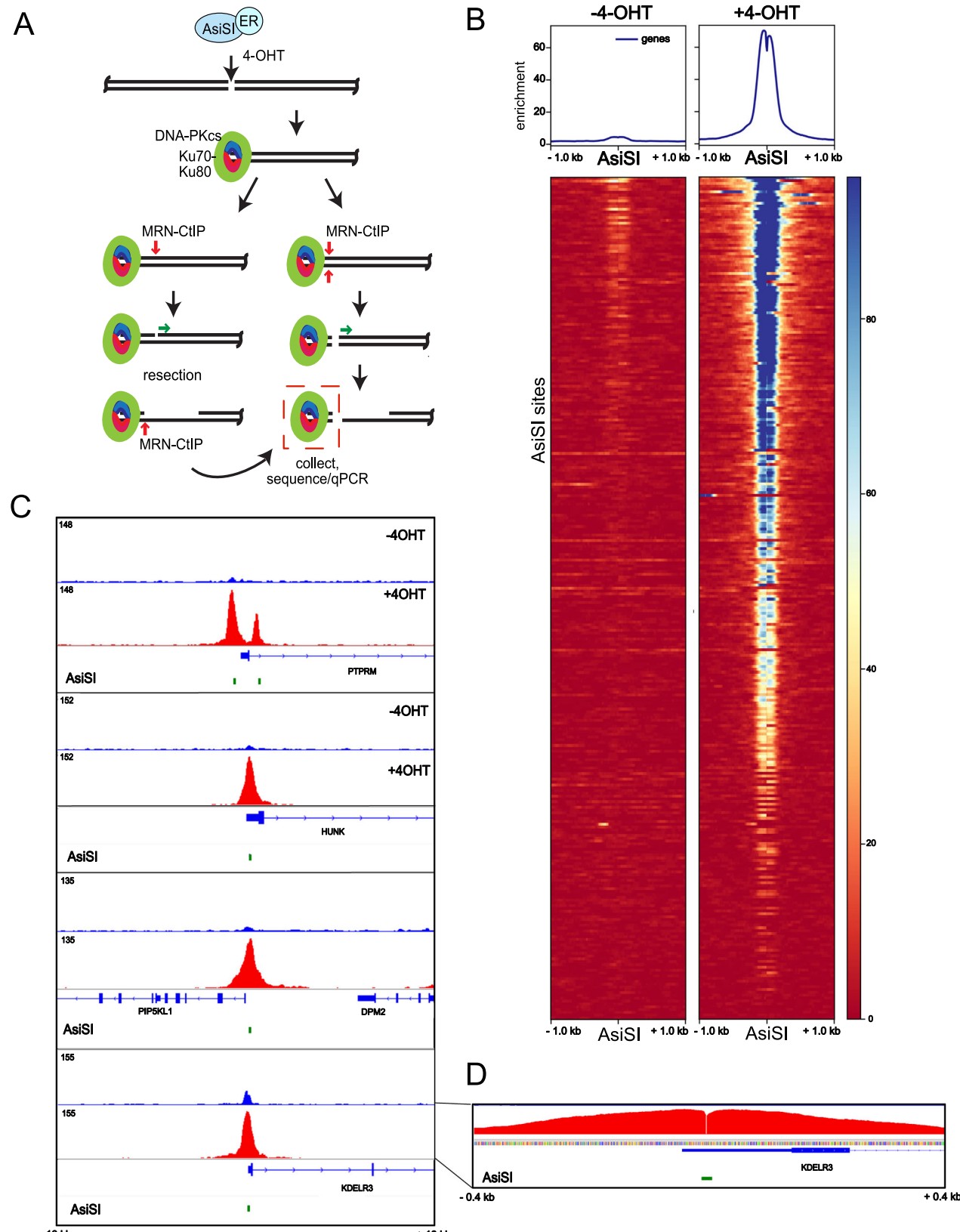

In the AsiSI DiVa system, a DSB site can be cleaved and repaired multiple times during the 4 h time window of 4-OHT addition if the site is repaired back to an intact AsiSI recognition sequence. In the absence of DNA-PKi, we noticed that there is detectable DNA-PK ChIP signal that extends across the central 2 nt of the AsiSI site, although the majority of the reads align on either side of the site, not crossing this boundary (Fig. 4D). The ChIP signal that extends across the 2 nt 3' overhang could be explained by DNA-PK being cross-linked to a site that is partially or fully repaired. To investigate this we performed a statistical analysis of the ChIP-seq signal in aligned reads using the top 50 most efficiently cut sites in order to determine the frequency of ChIP reads that do or do not extend across the central 2 nt sequence,

**Fig. 1 | DNA-PKcs-bound fragments are released from DSB sites in human cells. A** Diagram of DSB induction by ER-fused AsiSI, which translocates into the nucleus with 4-OHT addition. Model for end processing: DNA-PK binds to DSB ends, which are recognized by the MRN complex and CtIP and are processed by endonucleolytic and exonucleolytic nuclease cutting by Mre11. One of the products of this processing is hypothesized to be DNA-PKcs-bound fragments of DNA, released by processing of both strands of DNA simultaneously (red box). These products can be recovered through the GLASS-ChIP procedure and analyzed by sequencing. **B** Recovery of GLASS-ChIP products at the top 300 AsiSI cutting sites is shown, in the absence or presence of 4-OHT as indicated. U2OS cells were induced with 4-OHT for 4 h in the presence of DNA-PKi (NU7441). **C** Genome browser views of 4 AsiSI sites showing GLASS-ChIP recovery with 4-OHT addition. Green marks indicate locations of AsiSI sites. **D** High-resolution view of one GLASS-ChIP peak showing the absence of signal at the center of the AsiSI cut site.

considering only reads that are found within 5 nt of the AsiSI site (Fig. S3). These results show that, as expected, the vast majority of DNA-PK bound fragments from DNA-PKi-treated cells do not cross this central 2 nt gap, while the reads obtained in the absence of inhibitor show a significant number of reads that extend across this sequence.

## CtIP ChIP shows correlation with DNA-PKcs and extends outward from DSB sites

To determine if end processing factors align spatially with NHEJ factors at DSBs we identified sites of CtIP association with AsiSI-induced DNA breaks using CtIP ChIP-seq (Fig. 5). Using sonicated chromatin as the starting material, we observed substantial binding of CtIP around AsiSI sites ("CtIP PC") (Fig. 5A). Comparisons of this signal with DNA-PK pellet ChIP show that CtIP does not associate generally with promoter regions as observed with DNA-PKcs but appears specifically at DSB sites with 4-OHT addition (Fig. 5B). Released fragments were analyzed for CtIP association as well ("CtIP GC") but no signal was observed, thus CtIP does not appear to be associated with the DNA-PK-bound ends that are released from the chromatin.

CtIP ChIP signal at AsiSI DSB sites extends even further than DNA-PKcs, as far as several kb distal from the DSB, although the majority of the mapped reads are within 2.5 kb of the break site (Fig. 5A, B). This accumulated signal suggests an involvement of the protein not only in the initiating events close to the break that promote DNA end resection but also potentially in long-range resection activities.

Interestingly, we found substantial CtIP signal at the center of the AsiSI cut site, extending across the 2 nt gap that is formed at fragments containing the AsiSI-generated 3′ overhang (Fig. 5C), in contrast to the absence of ChIP signal observed at DNA-PK-bound fragments (Figs. 1D, 4D). The majority of mapped reads in CtIP pellet-ChIP data extend across this gap (Fig. S3). A statistical comparison of DNA-PK and CtIP mapped reads at AsiSI sites in comparison to previously reported BLESS signal[28] which labels DSBs[29] shows that the CtIP span ratio (proportion of mapped reads extending across the center 2 nt) is significantly higher than with DNA-PK ChIP in the absence of inhibitor and is comparable to that of LIG4. A similar analysis was performed with Mre11 ChIP products and Cas9-induced DSBs[30] which showed that the reads for Mre11 are primarily abutting the DSB site, not spanning, thus MRN and CtIP appear to have different patterns. The binding of DSB factors to sequences spanning the AsiSI site likely indicate situations where the factors remain bound for some time after the site has been repaired. This might be expected for enzymes involved in the final step of repair, such as LIG4 in NHEJ, but is an unexpected finding for CtIP which is thought to function primarily in the initiating stages of end processing.

Quantitative analysis of the accumulation of CtIP ChIP signal at AsiSI cut sites shows a strong correlation with patterns of DNA-PKcs GLASS-ChIP ($\rho = 0.69$), much higher than the correlation with chromatin-associated DNA-PKcs ($\rho = 0.24$) (Fig. 5D, E), consistent with our previous observations that CtIP collaborates with MRN and is essential for the endonuclease-mediated release of DNA-PK-bound ends[9]. The highest correlation with CtIP ChIP is observed with previously published Xrcc4 and Rad51 ChIP data[24] (Fig. 5F, G) with correlation coefficients 0.81 and 0.77, respectively. Previous work identified sites that were categorized as HR-prone or NHEJ-prone, depending primarily on the presence or absence of Rad51 at these sites, with Xrcc4 observed to be present at all cutting sites[24]. The HR-prone subset of

sites shows the highest levels of normalized Rad51 signal as well as CtIP signal, although the NHEJ-prone sites also exhibit high levels of these HR-associated factors (Fig. 5G).

## DNA-PKcs-bound GLASS-ChIP product efficiency aligns with Rad51 as well as NHEJ factors

Looking more broadly at correlations between DNA-PKcs recovery at AsiSI sites and other DSB-binding factors and chromatin marks, we find that the highest correlation observed with DNA-PKcs GLASS-ChIP efficiency is Rad51, followed by the DSB factors Xrcc4, Lig4, and CtIP (Fig. S4A). The pellet-ChIP signal also aligned closely with these DSB-associated factors, although less so compared to the released DNA-PKcs signal, and the DNA-PKcs pellet ChIP showed a much higher correlation with specific marks associated with an open chromatin state and transcription activation (H4K20me1, H2BK120Ub, and H3K36me3)[31–33]. Consistent with the close association between the efficiency of DNA-PKcs ChIP recovery and DSB-associated factors, we found that hierarchical clustering of the AsiSI genomic locations for all of the reported chromatin marks shows that DNA-PKcs (both GLASS-ChIP and pellet-ChIP) clusters together with CtIP, Rad51, 53BP1, Lig4, and Xrcc4 (Fig. S4B).

Some of the chromatin marks showing significant correlations with DNA-PKcs ChIP recovery are associated with S phase or mitosis[34,35] and thus may be correlated due to cell cycle specificity. To determine if the MRN cutting adjacent to DNA-PKcs is specific to certain cell cycle phases, we tested the efficiency of GLASS-ChIP in $G_0$, $G_1$, and $G_2$ phases by qPCR at two AsiSI sites PTPRM and PIP5KL1 (Fig. S4C). This analysis showed that GLASS-ChIP products are observed in all cell cycle phases at approximately 10% efficiency compared to DNA-PKcs pellet-CHIP fragments. A similar analysis with non-AsiSI DNA-PKcs binding sites H2AC21 and SNORD3A shows varying ratios depending on the gene, while the levels of DNA-PKcs bound at these sites increases substantially in $G_1$ and $G_2$ phases relative to $G_0$. This may be related to increased transcription of these genes since histone biosynthesis is cell cycle-regulated[36] and small nucleolar RNAs are tied to ribosomal RNA maturation which is also determined by cell cycle phase[37]. To confirm that resection of AsiSI-induced breaks actually occurs in all cell cycle phases, we measured resection using a qPCR method[38] (Fig. S5). These results confirmed that a low level of resection occurs in $G_0$ cells as measured by this assay, but is significantly higher in $G_1$ and $G_2$ cells. We also confirmed that exposure to DNA-PKi increases resection efficiency as we previously reported[39].

To quantitatively assess non-AsiSI sites throughout the genome for DNA-PKcs occupancy and other markers, the top 2500 DNA-PKcs-enriched genomic sites from the GLASS-ChIP dataset were analyzed for previously reported chromatin marks as well as for DSB-binding factors[23,24,28]. Binding for each factor or mark was ranked within the 2500 genomic locations and analyzed by unsupervised hierarchical clustering (Fig. S6). This analysis shows that approximately 20% of the highest efficiency DNA-PKcs-bound sites in U2OS cells are also associated with binding of DSB factors Lig4, Xrcc4, 53BP1, and Rad51, likely indicating the presence of naturally-occurring DSBs. Other chromatin marks are also closely associated with a subset of these sites, including γ-H2AX, H3K36me2, and H4S1p. The remaining 70% to 80% of the highest DNA-PKcs binding sites are associated with high levels of RNAPII but not with DSB-associated marks.

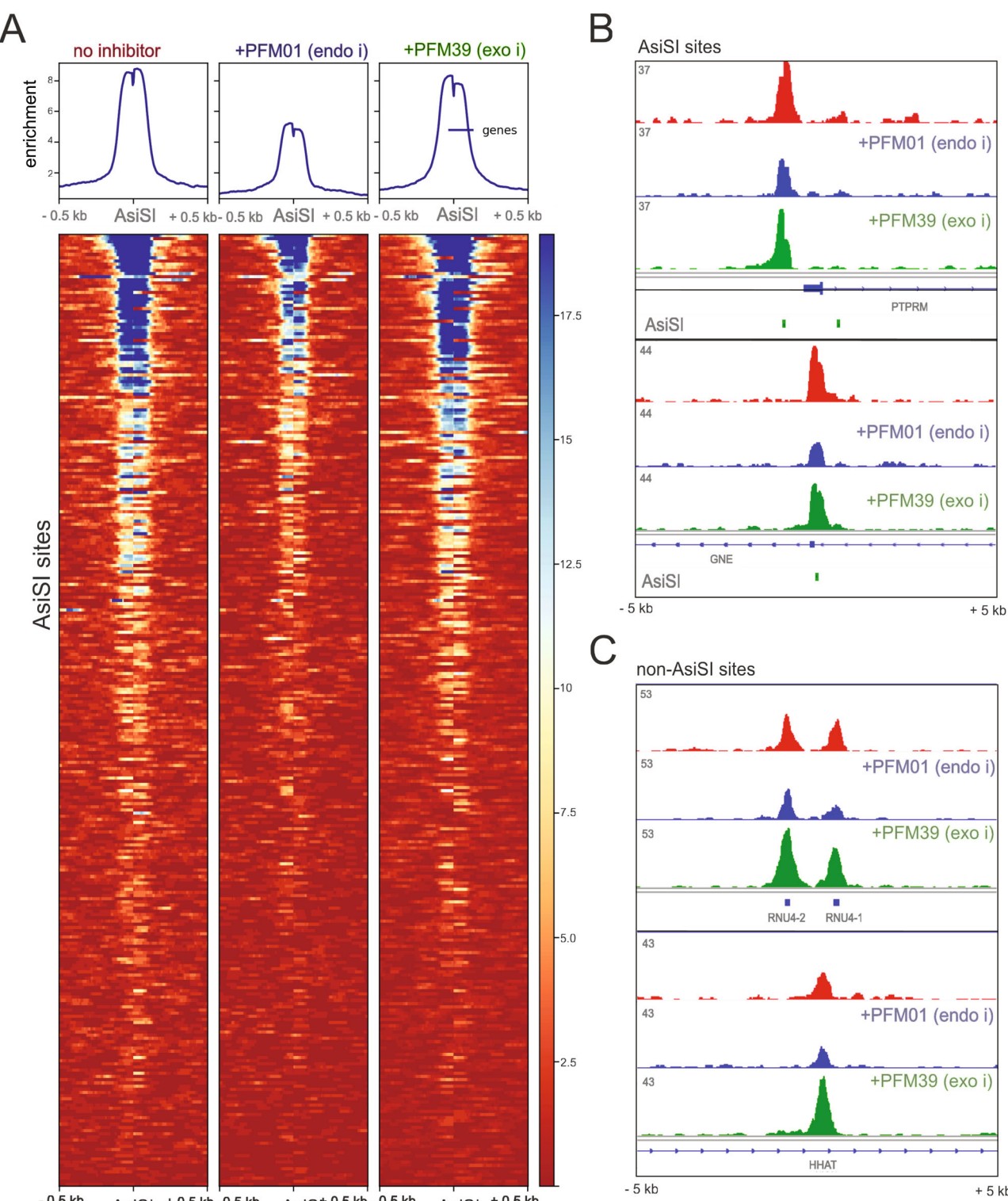

**Fig. 2 | Inhibition of Mre11 nuclease activity blocks release of DNA-PKcs-bound fragments. A** Recovery of GLASS-ChIP products at the top 300 AsiSI cutting sites is shown in the absence or presence of PFM01 (Mre11 endonuclease inhibitor), or PFM39 (Mre11 exonuclease inhibitor) as indicated. U2OS cells were induced with 4-OHT for 4 h in the presence of DNA-PKi (NU7441). **B** Genome browser views of 2 AsiSI sites showing GLASS-ChIP recovery with 4-OHT addition and inhibitors as indicated. Green marks indicate locations of AsiSI sites. **C** Genome browser views of non-AsiSI sites showing GLASS-ChIP recovery with 4-OHT addition and inhibitors as indicated.

## DNA-PKcs and Mre11 coincide at DSB sites before MRN cleavage product is formed

Although we have documented the binding and release of DNA-PKcs from chromatin with AsiSI induction, it is challenging to characterize the kinetics of these events with this system since the tamoxifen-induced translocation of estrogen receptor fusions occurs gradually over the course of several hours and the AsiSI enzyme also has the opportunity to cut religated sites multiple times[40]. Thus we turned to the recently developed Cas9-based system where a guide RNA containing caged nucleotides allows for Cas9 recognition of a target site,

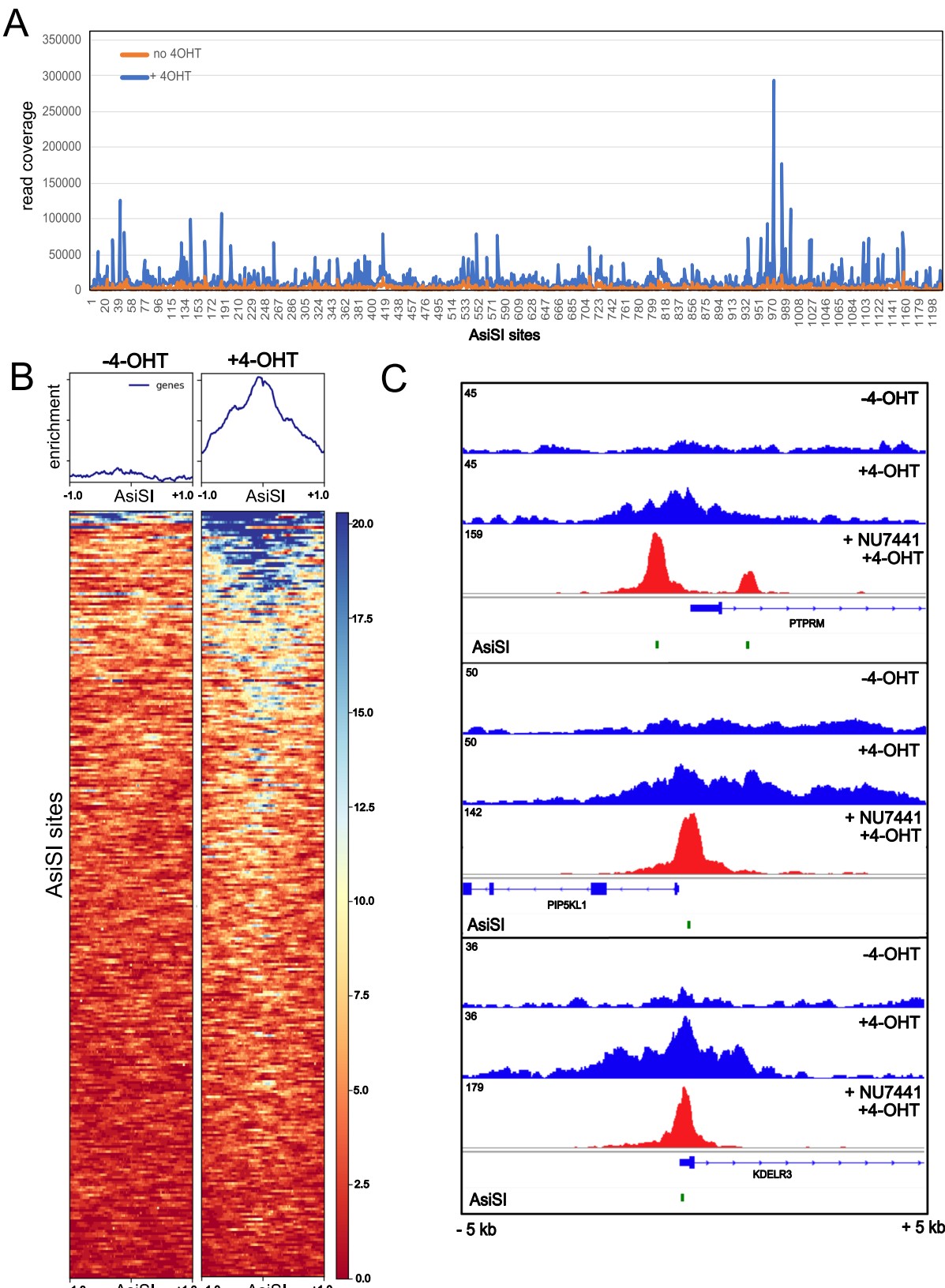

**Fig. 3 | Recovery of DNA-PKcs-bound fragments occurs in the absence of DNA-PKi. A** Summary of GLASS-ChIP enrichment (total read depth) at all 1211 AsiSI sites (includes 1 kb region upstream and downstream of site), in the absence of DNA-PKi. **B** Recovery of GLASS-ChIP products at the top 300 AsiSI cutting sites is shown, in the absence or presence of 4-OHT (4 h) as indicated. **C** Genome browser views of 3 AsiSI sites showing GLASS-ChIP recovery with 4-OHT addition and comparison with samples exposed to DNA-PKi. Green marks indicate locations of AsiSI sites.

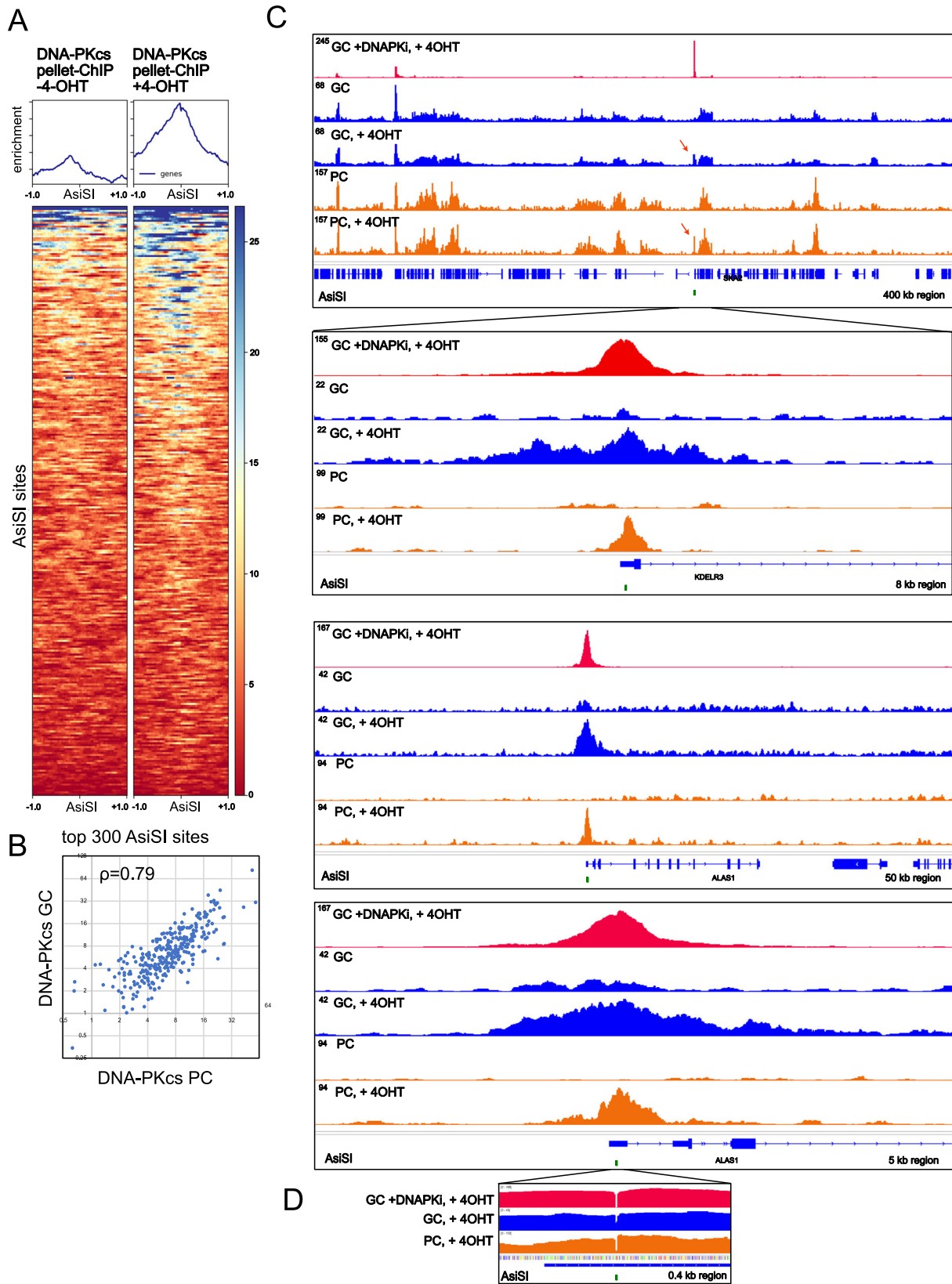

**Fig. 4 | Comparisons between DNA-PKcs released and chromatin-bound fragments in the absence of DNA-PKcs inhibition. A** Recovery of DNA-PKcs pellet-ChIP products at the top 300 AsiSI cutting sites is shown, in the absence or presence of 4-OHT (4 h) as indicated. **B** Enrichment for DNA-PKcs GLASS-ChIP versus pellet ChIP is shown for the 300 most efficiently cut AsiSI sites, with Pearson correlation coefficient as indicated. **C** Genome browser views of 2 AsiSI sites showing GLASS-ChIP and DNA-PKcs pellet-ChIP recovery with 4-OHT addition. Insets show higher resolution views. Green marks indicate locations of AsiSI sites. **D** High-resolution view of one peak showing signal at the center of the AsiSI cut site. Source data are provided as a Source data file.

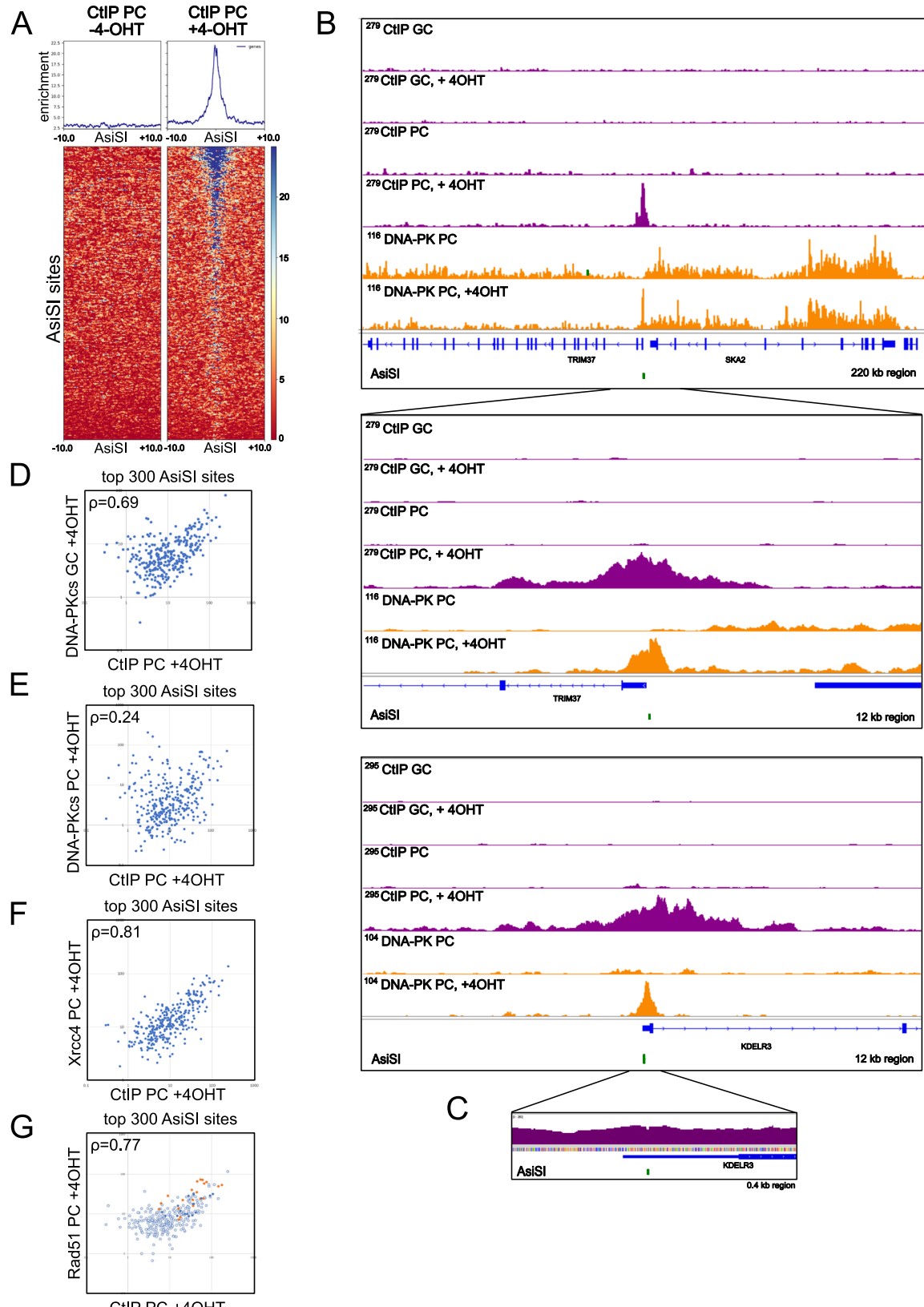

**Fig. 5 | CtIP ChIP patterns show extended binding around DSB site. A** Recovery of CtIP pellet-ChIP products at the top 300 AsiSI cutting sites is shown, in the absence or presence of 4-OHT (4 h) as indicated. **B** Genome browser views of 2 AsiSI sites showing CtIP ChIP signal (both released fragments "CtIP GC" and pellet CtIP ChIP "CtIP PC") as well as DNA-PKcs pellet-ChIP signal ("DNA-PK PC") with 4-OHT addition. Insets show higher resolution views. Green marks indicate locations of AsiSI sites. **C** High-resolution view of one CtIP ChIP peak. **D–G** Comparisons of accumulated ChIP signal at the top 300 AsiSI cut sites with CtIP ChIP, DNA-PK GLASS-ChIP, DNA-PK pellet-ChIP, Xrcc4 ChIP, and Rad51 ChIP with Pearson correlation coefficients as indicated. **G** includes designations of HR-prone sites (orange) and NHEJ-prone sites (black)[24]. Xrcc4 and Rad51 data from previously published datasets[24]. Source data are provided as a Source data file.

but does not undergo cleavage of the DNA until cells are exposed to light of an appropriate wavelength to remove the caged adduct (365 nm)[41]. This CRISPR system allows for high temporal resolution of events since Cas9 cleavage of the target site occurs within seconds of light exposure. Subsequent work using this light-activated system also generated several multi-target guide RNAs, each of which recognizes a large number of identical targets in the genome[30]. Here we used one of these multi-target guides (AluGG) that recognizes 126 copies of a short interspersed nuclear element.

With the light-activated Cas9 in HEK293T cells, we performed GLASS-ChIP of DNA-PKcs released from chromatin, in the presence of DNA-PKi (Fig. 6A). Analysis of all 126 target sites shows robust product accumulation at 60 min post light exposure, but no DNA-PKcs fragments at 15 min (Fig. 6B). In contrast, Mre11 ChIP performed under the same conditions shows accumulation of MRN at the 15 min time point (Fig. 6C, D), as reported previously in the absence of DNA-PKi[30]. Thus, the MRN complex is present at the DSB early after break induction but the DNA-PKcs product requires a longer occupancy at the break site.

We also examined the DNA-PKcs present in the chromatin at varying time points after light exposure and found that the protein is present at the target sites at an early time point (15 min, DNA-PKcs pellet-ChIP) (Fig. 7A). Comparison of Mre11 and DNA-PKcs pellet ChIP patterns show a similar accumulation at this time whereas the released GLASS-ChIP product is not present (Fig. 7B), suggesting that the MRN complex and DNA-PKcs occupy the same sites immediately following break induction and that the DNA-PKcs-bound product is released subsequent to this co-localization. Examination of the locations of ChIP products at the 15 and 60 min time points suggests that the chromatin-bound DNA-PKcs (pellet-ChIP, PC) is located at the break site while Mre11 is positioned adjacent to this, away from the break site (Fig. 7C). Genome browser views of two Cas9 target sites show examples of this coincident accumulation of DNA-PKcs and Mre11 at specific cut sites (Fig. 7D).

Quantitative analysis of the accumulation of Mre11 ChIP signal at Cas9 cut sites shows a strong correlation with patterns of DNA-PKcs GLASS-ChIP at the 60 min time point ($\rho = 0.81$), significantly higher than the 15 min time point when GLASS-ChIP signal is minimal ($\rho = 0.31$) (Fig. 7E). Correlations between Mre11 signal and DNA-PKcs pellet-ChIP are high at both time points ($\rho = 0.81$ and $0.87$, respectively) (Fig. 7F). These observations indicate that Mre11, similar to CtIP, associates with DSB sites with similar preferences compared to DNA-PKcs. The variation in signal is likely determined by the efficiency of cutting, as previously demonstrated with Cas9 ChIP[30], as well as epigenetic factors (Figs. S4, S6).

Lastly, we examined occupancy of DNA-PK and Mre11 relative to the Cas9 cut sites by calculating span ratio (Fig. S3). These results show that Mre11 is predominantly located on ChIP fragments adjacent to the break site, similar to previous analysis[30].

## Discussion

In this study we document the production of MRN-dependent fragments of DNA, bound by DNA-PK, and released from the genome at sites of DNA DSBs. Production of these fragments is sensitive to the endonuclease-specific Mre11 inhibitor PFM01. This result, combined with data from purified protein reconstitution experiments and single-molecule assays[9], suggests that MRN endonuclease activity processes DNA-PK-bound ends in genomic DNA to release short, double-stranded products.

From the in vitro assays we and others have performed[6,8–11,42], we know that a protein block is essential for MRN(X) endonucleolytic cutting at DSBs. DNA-PK is an excellent candidate for such a block because it binds tightly to ends through the Ku heterodimer. We also know from laser irradiation studies that it is one of the first complexes to arrive at a DSB, generally faster than Mre11 kinetics[43–45]. In vitro we observed both 5′ nicks as well as concurrent 5′ and 3′ cutting by MRN,

which generates a new DSB[8,9], with the nicking activity about 7 to 8-fold more efficient than double-strand cuts. The position of the endonuclease cleavage in vitro with DNA-PK, MRN, and CtIP present is approximately 45 nt from the end of the 5′ strand. In cells, however, with DNA-PKi present, we find that the fragments are significantly larger—on average 150–180 nt from the AsiSI cut site. We do not know the reasons for this difference in cut location but speculate that other end-binding factors present in cells that associate with DNA-PK likely contribute to the size of the end-bound complex. Mass spectrometry characterization of proteins bound to the DNA-PK-associated fragments show the MRN complex and several other DNA repair factors including PARP, Xrcc1, FEN1, and Mdc1 (Fig. S7). In addition, a large number of RNA splicing and transcription-associated proteins are enriched on these ends.

When DNA-PK is inhibited in vitro, the complex remains bound very stably to DNA ends[22]. This nearly irreversible binding is due to the fact that DNA-PKcs autophosphorylation is required to promote release from DNA[19–21] and that the phosphorylation of Ku70 by DNA-PKcs also promotes release of Ku[46], so when these events are blocked, DNA-PK is immobilized. In vitro, DNA-PKi strongly promotes MRN nuclease activity[9], similar to the robust release of DNA-PK-bound fragments in the presence of DNA-PKi observed here in cells.

In the absence of DNA-PKi, induction of DSBs by AsiSI produces DNA-PK-bound fragments that map to sites directly adjacent to the DSB location but also appear farther away—up to 1 kb or more depending on the site. It is not clear what generates this distribution. One possibility is that multiple binding and cutting events occur over the time period of AsiSI translocation into the nucleus. Alternatively, there could be movement of Ku on DNA such that sliding and stacking of DNA-PK occurs, ultimately generating released products that extend far from the original site of the DSB. We favor the latter model, based on early experiments with DNA-PK in human cell nuclear extracts that suggested movement of the complex inward from DNA ends in a manner dependent on DNA-PK catalytic activity[47]. This contrasts with the ATP-independent sliding of Ku alone[48,49].

Interestingly, the extent of DNA-PKcs spreading observed using GLASS-ChIPseq in the absence of DNA-PKi is more extensive than observed in standard pellet-ChIP using the same conditions. Interpreted in light of the sliding model, this result would suggest that MRN processing of DNA-PK-bound fragments occurs more efficiently at sites of internalized DNA-PK compared to all DNA-PK-bound sites.

We also observed here that CtIP accumulates at distal locations relative to DSB sites in the absence of DNA-PK inhibition. This could be interpreted as evidence for CtIP association with resection intermediates along the entire resection tract, not just in the immediate vicinity of the break. If so, the role of CtIP in promoting end processing might not be limited exclusively to the initial phase of short-range resection. Previous work has shown that CtIP recruitment to DSBs occurs through direct DNA binding as well as interactions with Nbs1, Brca1, and CTCF[50–54]. CtIP has also recently been shown to bridge DNA ends in vitro[55], which may be related to its association at distal sites shown here.

With the AsiSI-ER DivA system, characterization of the kinetics of DNA-PK removal is challenging due to the extended period of AsiSI translocation into the nucleus. The light-activated Cas9 system[30,41], however, allows for precise control of the cleavage step and therefore gives us the opportunity to look at early time points following break induction. Here we found that Mre11 and DNA-PK are both found on chromatin adjacent to the Cas9 cut site at an early time following light exposure (15 min). At this time, released DNA-PK-bound fragments (GLASS-ChIP) are not observed, but production of these fragments is clearly seen at a later time point (60 min). Thus, we infer that MRN and DNA-PK likely occupy the same ends for some time after break induction (overlapping with the 15 min time point) and that MRN processing of these ends occurs after this

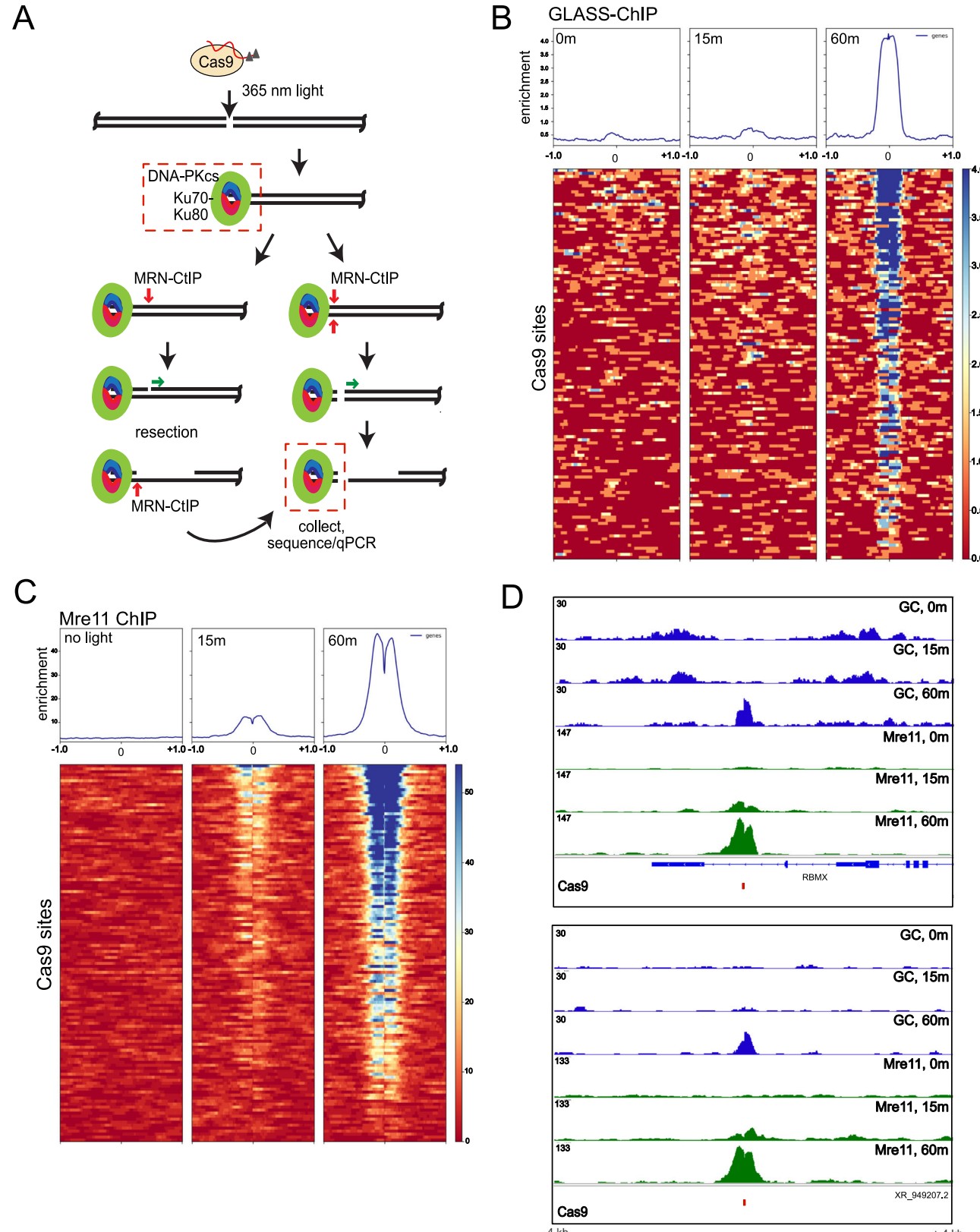

**Fig. 6 | GLASS-ChIP recovery of DNA-PKcs-bound fragments at light-activated Cas9 target sites. A** Diagram of DSB induction by light-activated Cas9. Model for end processing as in Fig. 1A. **B** Recovery of GLASS-ChIP DNA-PKcs products at the 126 target sites for the AluGG guide RNA are shown, without 365 nm light activation (0 min) or 15 or 60 min following light exposure as indicated, in the presence of DNA-PKi (NU7441). **C** Recovery of Mre11 ChIP products at the 126 target sites for the

AluGG guide RNA are shown, without 365 nm light activation (0 min) or 15 or 60 min following light exposure as indicated, in the presence of DNA-PKi (NU7441). **D** Genome browser views of 2 Cas9 target sites showing GLASS-ChIP and Mre11 recovery at 0, 15 min, or 60 min post light exposure. Red marks indicate locations of Cas9 target sites.

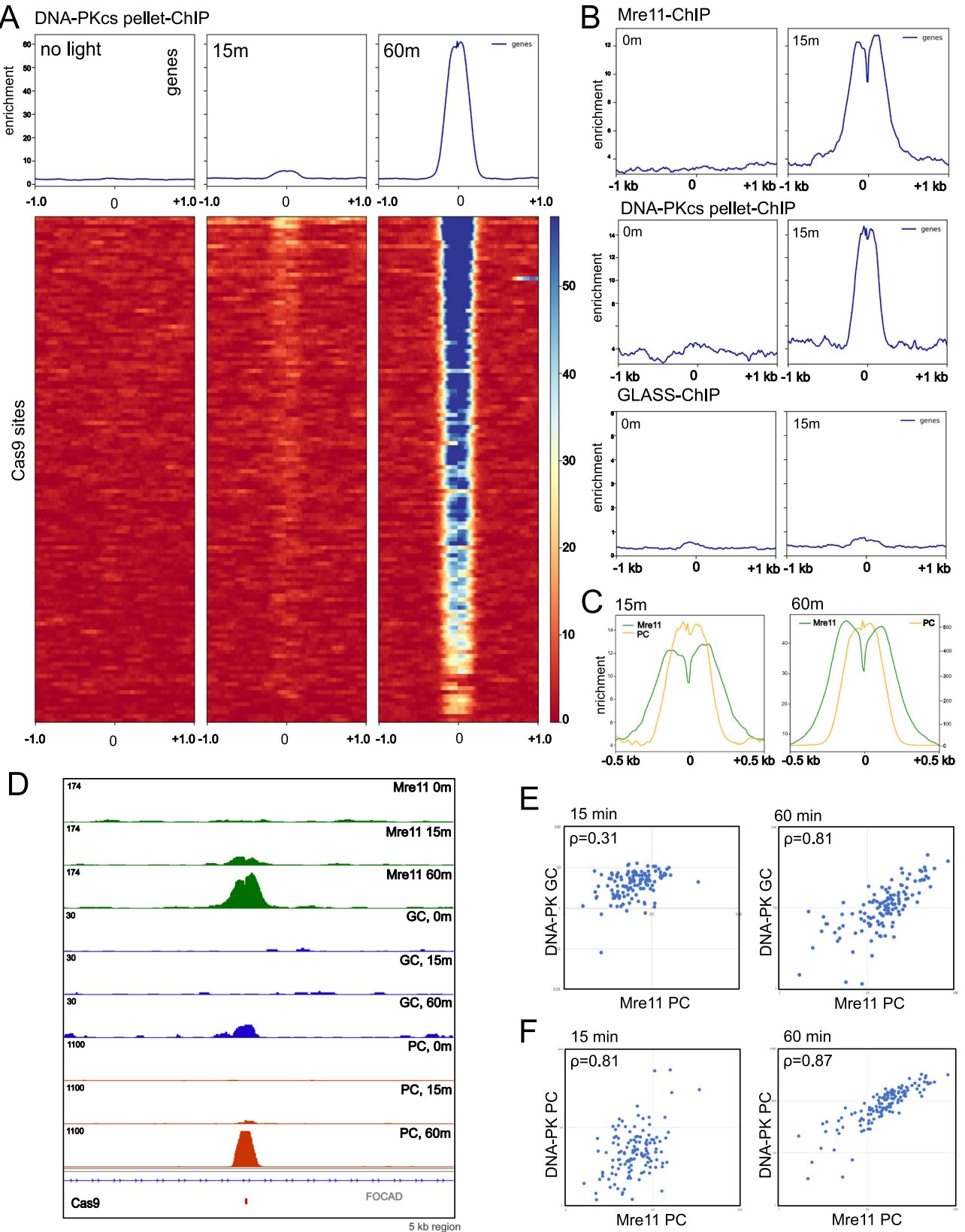

simultaneous binding period. From the pattern of ChIP occupancy we observed, DNA-PK is bound at the break site while MRN is located adjacent to this, away from the end. This interpretation is also supported by our previous single-molecule observations where MRN colocalization with DNA-PK on DNA ends was frequently observed and preceded Mre11 nuclease-dependent loss of both DNA-PK and MRN

from the ends[9]. In these experiments, we also observed MRN complexes internal to DNA-PK-bound ends before release. We do not know what the trigger is for the processing event, but it is likely a conformational change inherent to the MRN, CtIP, and DNA-PK complexes considering that the in vitro experiments utilized purified recombinant proteins.

**Fig. 7 | DNA-PKcs and Mre11 bind to DSB site before DNA-PKcs is released.**
**A** Recovery of DNA-PKcs ChIP products (pellet ChIP) at the 126 target sites for the AluGG guide RNA are shown, without 365 nm light activation (0 min) or 15 or 60 min following light exposure as indicated, in the presence of DNA-PKi (NU7441). **B** Comparison of Mre11 ChIP, DNA-PKcs pellet ChIP, and GLASS-ChIP products at the 0 min and 15 min time points, using equivalent enrichment scales. **C** View of Mre11 ChIP and DNA-PKcs pellet ChIP (PC) signal at 15 min and 60 min after light exposure. Note separate Y axis for DNA-PK pellet ChIP at 60 min. **D** Genome browser views of a Cas9 target site showing GLASS-ChIP, DNA-PKcs pellet-ChIP, and

Mre11 ChIP at 0, 15 min, or 60 min post light exposure. Red mark indicates locations of Cas9 target sites. **E** Comparison of accumulated Mre11 pellet ChIP signal at the 126 Cas9 target sites compared to DNA-PK GLASS-ChIP signal at 15 min and 60 min post light exposure with Pearson correlation coefficient as indicated.
**F** Comparisons of accumulated Mre11 pellet ChIP signal at the 126 Cas9 target sites compared to DNA-PK pellet ChIP signal at 15 and 60 min post light exposure with Pearson correlation coefficients as indicated. Source data are provided as a Source data file.

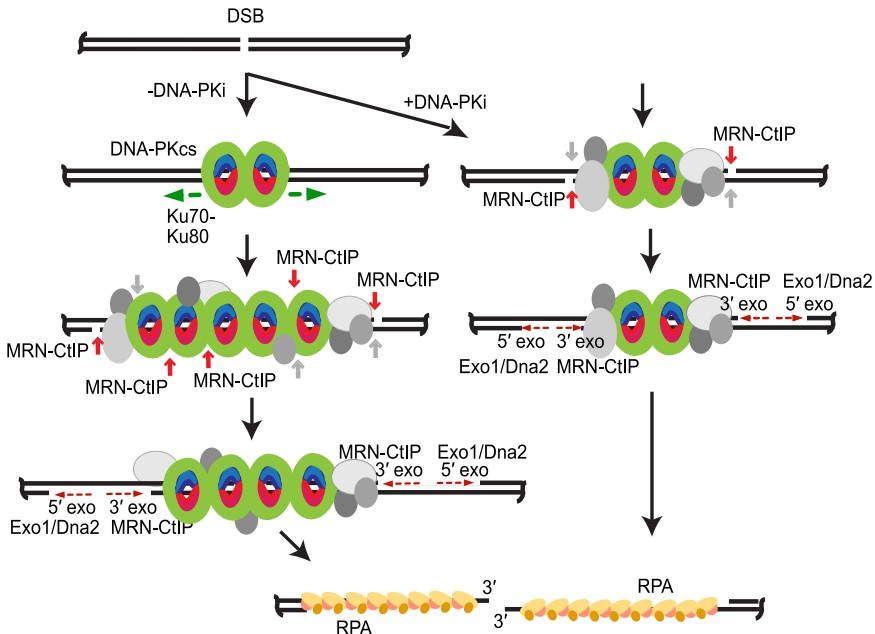

**Fig. 8 | Working model for DNA-PK and MRN dynamics at DSB sites.** After DSB formation, DNA-PK (DNA-PKcs green, Ku red/blue) binds to DSBs. In the presence of DNA-PKi (right), DNA-PK remains tightly bound at the DSB site, together with accessory proteins (gray). MRN nicks on the 5′ strands occur (red) and, less frequently, simultaneous nicks on the 3′ strands occur (gray). Simultaneous nicks on both strands give rise to GLASS-ChIP products. Further processing on the 5′ strands by MRN (3′ to 5′ exonuclease, red dashed arrow towards DSB) and by Exo1 or Dna2 (5′ to 3′ exonuclease, red dashed arrow away from DSB) extend nicks into gaps of

varying length. Ultimately, gapped intermediates are bound by RPA (yellow/orange) and DNA-PK is removed. In the absence of DNA-PKi (left), DNA-PK slides inward (green arrows), allowing more DNA-PK molecules as well as accessory factors to bind. Alternative model: multiple rounds of cutting and re-binding allow for DNA-PK movement inward (not shown). MRN cutting on 5′ strands and less efficient cutting on 3′ strands occurs as described above, followed by exonucleolytic expansion of the gap.

The recovery of DNA-PKcs-bound DNA fragments genome-wide provides a global view of binding preferences. Correlation analysis of DNA-PKcs binding efficiency with other DNA repair proteins, chromatin marks, and transcription-related factors shows that the binding is most tightly correlated with RNAPII. DNA-PKcs has been observed to associate physically and functionally with RNAPII previously and in some cases was observed to be essential for transcription activation[26,27,56,57]. The exact role of DNA-PK in this context, however, has not been defined. Here we see that approximately 20% of the most efficiently bound DNA-PK sites in the genome that are coincident with RNAPII appear to be DSB sites, on the basis of coincident Xrcc4, Lig4, and 53BP1 signals. Considering that DSB formation has been shown to be essential for transcription in some contexts, it is possible that the DSB machinery is regulating transcription at these genomic locations. The remainder of the DNA-PKcs genomic binding sites show RNAPII but are devoid of other DSB factors. Recent work also shows that MRN occupancy in the genome is strongly correlated with RNAPII abundance[58] and that CtIP association with RNAPII at DSB sites helps to reactivate transcription after DNA damage at transcriptionally active sites[59].

Correlation analysis of DNA-PKcs binding in comparison with previously published U2OS DivA datasets from the Legube laboratory

also shows that GLASS-ChIP recovery within the most efficiently cut AsiSI sites shows the highest correlation with Rad51 and other DSB markers, including Xrcc4, Lig4, and 53BP1. This is consistent with expectations, considering that MRN processing is required for subsequent long-range resection and Rad51 loading. Hierarchical clustering of all of the AsiSI sites shows these preferences clearly, with the most efficiently cut sites clustering together and binding all of the DSB-specific factors. From this analysis, there do not appear to be separate groups of DSBs that bind to Rad51 or the NHEJ-related factors exclusively but all of these factors associate with the sites that are the most efficiently cut.

From our results in this work as well as our previous in vitro studies, we propose that DNA-PK plays an essential role in DSB processing by acting as a protein block to stimulate MRN end processing (Fig. 8). Most of this processing is likely in the form of 5′ strand nicks, based on ensemble reconstitution assays with purified proteins, while approximately 10 to 15% of the events lead to double-strand processing, releasing a DNA-PK-bound fragment. Overall this presents a very different view of "pathway choice" compared to the canonical model by positing that this must be a sequential series of events, initiated by DNA-PK first binding to DNA ends. Homologous recombination factors such as MRN are thus dependent on and collaborating with NHEJ

factors, an "entwined" rather than competitive scenario also predicted by mechanistic modeling of previously published data[60]. In future work, it will be important to define the rate-limiting step(s) for Mre11 cleavage and to better understand the role of DNA-PK internalization in promotion of MRN activity to elucidate the details of this collaboration with greater resolution.

## Methods

### Cell culture
Human U2OS cells were grown in DMEM with high glucose, L-glutamine, and sodium pyruvate and 10% fetal bovine serum, with 1X PEN/STREP. Recombinant AsiSI was expressed via the DivA system as previously described[15].

### GLASS-ChIP assay
Human U2OS cells with inducible AsiSI[15] were grown to 50–60% confluency in 150 mm dishes and treated with 600 nM 4-OHT for 4 h at 37 °C. When used, 10 μM NU7441 was added 30 min prior to 4-OHT addition. After 4-OHT treatment, cells were fixed with 1% formaldehyde for 7 min at RT with gentle rotation. Cross-linking was stopped by addition of 125 mM glycine for 5 min, washed twice with cold PBS, harvested, flash-frozen in liquid nitrogen, and stored at −80 °C.

For the GLASS-ChIP assay, we modified a standard protocol (Abcam) with a gentle lysis procedure and minimal, low-level sonication to rupture cells without extensive DNA damage as previously described[16]. The formaldehyde fixed cells were thawed at RT for 5 min, resuspended in RIPA buffer (50 mM Tris-HCl pH 8.0, 150 mM NaCl, 2 mM EDTA pH 8.0, 1% NP-40, 0.5% Sodium Deoxycholate, 0.1% SDS) with 1x protease inhibitors (Pierce #A32955) and sonicated using a Cell Ruptor at low power, for 10 s followed by 10 pulses after a 20 s interval. Cell lysates were then centrifuged at 3000 rpm for 3 min at RT to remove the bulk of chromatin. The supernatant was then incubated with 1.6 μg of anti-DNA-PKcs pS2056 antibodies (Abcam 124918) overnight at 4 °C, followed by incubation with 25 μl Protein A/G magnetic beads (Pierce) at RT for 2 h. Beads were then washed sequentially once in low salt wash buffer (0.1% SDS, 1% Triton X-100, 2 mM EDTA, 20 mM Tris-HCl pH 8.0, 150 mM NaCl), once in high salt wash buffer (0.1% SDS, 1% Triton X-100, 2 mM EDTA, 20 mM Tris-HCl pH 8.0, 500 mM NaCl), once in LiCl wash buffer (0.25 M LiCl, 1% NP-40, 1% Sodium Deoxycholate, 1 mM EDTA, 10 mM Tris-HCl pH 8.0). Beads were then resuspended in TE buffer (10 mM Tris pH 8.0, 0.1 mM EDTA) and transferred to a fresh tube and finally eluted with 100 μl elution buffer (1% SDS, 100 mM NaHCO₃). Cross-links were reverted for the elutions (65 °C for 24 h) and DNA was purified with a Qiagen Nucleotide Clean up kit. DNA fragments <300 bp from ChIP elutions were separated by pulling down larger fragments using paramagnetic Ampure XP beads: 65 μl of Ampure XP beads were added to the uncrosslinked ChIP elution and mixed thoroughly. After 10 min incubation at RT, beads were isolated using a magnet. The supernatant was collected and 25 μl of fresh Ampure XP beads were added and mixed thoroughly. After 10 min at RT, beads were separated and the supernatant was purified using Qiagen Nucleotide Clean up kit. At the final step DNA was eluted in 60 μl of elution buffer (TE). To monitor the Mre11 nuclease dependence, we treated the human U2OS cells expressing inducible AsiSI with 10 μM NU7441, 100 μM PFM01 or PFM39 and 600 nM 4-OHT simultaneously for 1 h at 37 °C before harvesting cells. For CtIP GLASS ChIP experiments, methods were identical except that CtIP antibody (Active Motif, 61141) was used.

### Pellet ChIP
For DNA-PKcs pellet ChIP, the protocol for GLASS-ChIP was followed through the step where chromatin is isolated. The chromatin fraction was then resuspended in 2.2 ml RIPA buffer and was fragmented using a Diagenode Bioruptor on high setting for 30 min with 10 s on, 10 s off.

After removal of debris by centrifuging at 800 × g for 3 min, the supernatant was incubated with 1.6 μg of anti-DNA-PKcs pS2056 antibodies (Abcam 124918) overnight at 4 °C, followed by incubation with 25 μl Protein A/G magnetic beads (Pierce) at RT for 2 h. The rest of the procedure was identical to GLASS-ChIP above except that the AMPure bead size selection was not performed. For CtIP pellet ChIP experiments, methods were identical except that CtIP antibody (Active Motif, 61141) was used.

### Sequencing library preparation
For both GLASS-ChIP and pellet-ChIP sequencing libraries, the eluted DNA was used to make sequencing libraries using the NEBNext Ultra or Ultra II DNA Library Prep Kit for Illumina (NEB) with NEBNext Multiplex dual index primers with 12 amplification cycles and 2 additional AMPure XP clean-up steps at 0.8X. Libraries were sequenced by the UT Genomic Sequencing and Analysis Facility (RRID:SCR_021713) using a NovaSeq SP platform with PE150 runs.

### Data analysis
Raw data was pre-processed with BedTools fastp[61], mapped with BWA-MEM[62], duplicates were removed with SAMTools RmDup[63], and peaks called with MACS2[64]. For Glass-ChIP, no-antibody sample libraries were used as controls whereas for pellet-ChIP, input DNA libraries were used as controls. Output bedgraph files were processed further using MACS2's bdgcmp command with the ppois flag to remove the input influence from the ChIP bedgraph treatment files. Finally, bedgraph output was read depth normalized to make files equivalent in total signal.

Correlation analysis of GLASS-ChIP with previously published datasets[23,24,28] was performed using deepTools MultiBigWigSummary on MACS2 treatment output files with either a bed file of the top 300 AsiSI sites or with the top 2500 sites identified by MACS2 narrowpeak output, followed by conversion to rankings and hierarchical clustering of rankings for each factor using the Euclidean method with average linkage and visualization with Heatmapper[65].

### Electroporation and light activation of Cas9 RNP
In order to assemble Cas9-gRNA RNP complex, 2 μL of 100 μM caged AluGG crRNA (Bio-Synthesis)[30] was mixed with 2 μL of 100 μM tracrRNA (Integrated DNA Technologies) and heated to 95 °C for 3 min in a thermocycler. The cr:tracrRNA was then allowed to cool on benchtop for 5 min. Afterward, 3 μL of 10 μg/μL (~66 μM) of purified Cas9 and 8 μL of dialysis buffer (20 mM HEPES pH 7.5, and 500 mM KCl, 20% glycerol) was added to the annealed 4 μL 50 μM cr:tracrRNA for a total of 15 μL, and was thoroughly mixed by pipetting. This solution was incubated for 20 min at room temperature to allow for RNP formation.

HEK293T cells were maintained to a confluency of ~90% prior to electroporation. 12 million cells were trypsinized with 5 min incubation in the incubator. Trypsin was quenched using 1:1 of complete DMEM and cells were then harvested and centrifuged (3 min, 200 × g). The supernatant was removed, and cells were washed with 1 mL PBS (resuspend pellet, centrifuge with same settings as above). After the PBS wash, cells were resuspended in 90 μL of nucleofection solution (16.2 μL of Supplement solution mixed with 73.8 μL of SF solution from SF Cell Line 4D-Nucleofector™ X Kit L) (Lonza), transferred to the 15 μL RNP solution; and 2 μL of Cas9 Electroporation Enhancer (Integrated DNA Technologies) was added. The final solution (~125 μL) was gently mixed and transferred to a 100 μL cuvette (Lonza). Electroporation was then performed according to the manufacturer's instructions on the 4D-Nucleofector™ Core Unit (Lonza) using code CA-189. A total of 400 μL of DMEM complete was used to completely transfer the cells out of the cuvette, before plating to culture wells pre-coated with 1:100 collagen. Cells were incubated with caged Cas9 RNP for 12 h before light activation. NU-7441 DNA-PK inhibitor was added at 10 μM final

concentration 1 h before light exposure. For Cas9 photo-activation, cells were exposed to 1 min of 365 nm light exposure from a handheld blacklight (Amazon https://www.amazon.com/JAXMAN-Ultraviolet-365nm-Detector-Flashlight/dp/B06XW7S1CS/). Typically, 6 flashlights were used at once. Samples were harvested without light exposure (0 min), or 15 m and 1 h after light exposure.

## Mre11 ChIP

A minimum of 4 million cells were used for each ChIP reaction. For Mre11 ChIP measurements cells were washed once with room temperature PBS, then scrapped off the plate with 10 mL DMEM and transferred to 15 mL falcon tubes. 721 μL of 16% formaldehyde (methanol-free) was added, and the fixation reaction was incubated for 10 min at room temperature with rotation. 750 μL of 2 M glycine was added to quench the formaldehyde, followed by a 3 min incubation with rotation. Cells were spun down at 1200 × $g$ at 4 °C for 3 min, then washed twice with ice-cold PBS. The supernatant was removed, and the cross-linked cell pellet was resuspended in 4 mL lysis buffer LB1 (50 mM HEPES, 140 mM NaCl, 1 mM EDTA, 10% glycerol, 0.5% Igepal CA-630, 0.25% Triton X-100, pH to 7.5 using KOH, add 1x protease inhibitor right before use) for 10 min at 4 °C, then spun down 2000 × $g$ at 4 °C for 3 min. The supernatant was removed and cells were resuspended in 4 mL LB2 (10 mM Tris-HCl pH 8, 200 mM NaCl, 1 mM EDTA, 0.5 mM EGTA, pH to 8.0 using HCl, add 1x protease inhibitor right before use) for 5 min at 4 °C, spun down with the same protocol. The supernatant was removed, and cells were then resuspended in 1.5 mL LB3 (10 mM Tris-HCl pH 8, 100 mM NaCl, 1 mM EDTA, 0.5 mM EGTA, 0.1% Na-Deoxycholate, 0.5% N-lauroylsarcosine, pH to 8.0 using HCl, add 1x protease inhibitor right before use) and transferred to 2 mL tubes. Sonication was performed using a Fisher 150E Sonic Dismembrator with settings: 50% amplitude, 30 s ON, 30 s OFF for 12 min total time. The sonicated sample was spun down (20,000 × $g$ at 4 °C for 10 min), and the supernatant was transferred to a 5 mL tube. 1.5 mL of LB3 (with no protease inhibitor) and 300 μL of 10% Triton X-100 were added, and the entire solution was well mixed by gentle inversion. Beads pre-loaded with antibody (Novus NB100-142) were prepared before cell harvesting. 50 μL Protein A beads (Thermo Fisher) were used per IP and transferred to a 2 mL tube on a magnetic stand. Beads were washed twice with blocking buffer BB (0.5% BSA in PBS), then resuspended in 100 μL BB per IP. 3 μL of Mre11 antibody per IP (MRE11– Novus NB100-142) was added and the mixture placed on rotator for 1-2 h. Right before IP, the 2 mL tube was placed on a magnetic rack and washed 3x with BB, before resuspending in 50 μL BB per IP. 50 μL of beads in BB were transferred to each IP and placed in 4 °C rotator for 6+ hours.

ChIP samples were transferred to a 1.5 mL LoBind tube on a magnetic stand, washed 6x with 1 mL RIPA buffer (50 mM HEPES, 500 mM LiCl, 1 mM EDTA, 1% Igepal CA-630, 0.7% Na-Deoxycholate, pH to 7.5 using KOH), then washed 1x with 1 mL TBE buffer (20 mM Tris-HCl pH 7.5, 150 mM NaCl). The liquid was removed, and the beads containing ChIP'd DNA were eluted in 50 μL elution buffer EB (50 mM Tris-HCl pH 8.0, 10 mM EDTA, 1% SDS) and incubated 65 °C for 6+ hours to remove cross-links. 40 μL of TE buffer was mixed to dilute the SDS, followed by 2 μL of 20 mg/mL RNaseA (New England BioLabs) for 30 min at 37 °C. 4 μL of 20 mg/mL Proteinase K (New England BioLabs) was added and incubated for 1 h at 55 °C. The genomic DNA was column purified (Qiagen) and eluted in 41 μL nuclease-free water.

For preparation of sequencing libraries, end-repair/A-tailing was performed on 17 μL of ChIPed DNA using NEBNext® Ultra™ II End Repair/dA-Tailing Module (New England BioLabs), followed by adapter ligation using T4 DNA Ligase (New England BioLabs). Libraries were amplified with 13 cycles of PCR using single-indexed primers.

For MRE11 ChIP-Seq after Cas9 light activation, ChIP'ed DNA samples were pooled, quantified with QuBit (Thermo), Bioanalyzer (Agilent) and qPCR (BioRad), then sequenced on a NextSeq 500

(Illumina) using high-output paired 2×50 bp reads. Reads were demultiplexed after sequencing using bcl2fastq. Paired-end reads were aligned to hg38 using bowtie2. Samtools[63] was used to filter for mapping quality >=25, remove singleton reads, convert to BAM format, remove potential PCR duplicates, and index reads. AsiSI files are processed with Hg19 genome build; Cas9 light-activated data processed with Hg38.

## Analysis of span ratios in bam files from ChIP-seq data

ChIP and BLESS BAM files were converted to BED format using bedtools BamToBed. Reads within 5 nt of each AsiSI cut site were labeled as spanning or not spanning depending on proximity to the break. Investigating the BAM files on IGV and given the AsiSI consensus sequence GCGAT/CGC, the DSB was determined as the AT region of the string. Reads that were <=5 nt adjacent to the AT region, but never crossing, were labeled as not spanning. Reads that had any overlap with the AT region were labeled spanning. Spanning and not spanning reads were counted for each AsiSI cut site for ChIP experiment and the BLESS experiment. The spanning ratio for each AsiSI cut site for each experiment was calculated as (#reads that span)/(#reads that span + #reads that do not span). Bootstrapping was performed to pairwise compare experiment spanning ratio histograms. Briefly, two ChIP experiments' spanning ratio vectors were merged and randomly split into two equally sized partitions. Partitions were converted into histograms, then converted into cumulative distribution functions (CDFs). The Kolmogorov–Smirnov test statistic was measured (maximum distance between the two partition CDFs) and added to an array. 999 additional random partitions and KS statistics were performed for a total of 1000 bootstraps. The ground truth KS statistic was then measured for the two real ChIP experiments and compared to the saved bootstrapped values on a histogram to determine if the two spanning ratio distributions were significantly different.

## Resection

Resection of 5′ strands at AsiSI breaks was performed using a qPCR-based method as previously described[38,66]. Briefly, after 4 h treatment with 600 nM 4-Hydroxytamoxifen (4-OHT), cells were trypsinized and resuspended in 0.6% low-gelling-temperature agarose at a final concentration of 2 × 10⁶ cells/ml. The agar balls with cells were used for genomic DNA extraction and DNA end resection at selected DSB sites were quantitated by quantitative PCR (qPCR).

## Cell cycle synchronization

Cells were synchronized in $G_0$ by serum starvation for 3 days. $G_1$ cells were obtained by addition of serum to $G_0$ cells for 8 h followed by addition of 4-OHT for 4 h before harvesting for ChIP or resection assays. $G_2$ cells were obtained by growing cells in low-dose (2 μg/ml) aphidicolin overnight to synchronize cells at the $G_1$/S phase boundary. The aphidicolin was removed, cells were grown for 6 h followed by addition of 4-OHT for 4 h before harvesting for ChIP or resection assays.

## Quantitative PCR

qPCR monitoring of GLASS-ChIP and pellet-ChIP libraries was performed using primers for 4 different AsiSI sites (Fig. S8). %DNA was calculated using equation $2^{(Ct(input)-Ct(test))} \times 100\%$. DNA values obtained for IP in absence of antibodies were subtracted from the values obtained in presence of antibody and used for plotting the graphs in Fig. 5.

## Mass spectrometry

GLASS-ChIP was performed as described above, using no antibody samples as controls, in two biological replicates, except that the cells were cross-linked with formaldehyde for only 1 min. Samples eluted from the Protein A/G beads were processed through a modified Filter-

Assisted Sample Preparation protocol as previously described[67]. Protein identification by LC-MS/MS was provided by the University of Texas at Austin Proteomics Facility on an Orbitrap Fusion following previously published procedures[68]. Raw files were analyzed using label-free quantification with Proteome Discoverer 2.2. embedded with SEQUEST (Thermo Scientific) using the Uniprot Human reference proteome database. The search parameters used were as follows: two missed cleavages on a trypsin digest were permitted, no fixed modifications. Peptide identifications were filtered using Percolator, where a false discovery rate (FDR) of 1% ($q < 0.01$) was applied. Results were further refined by two additional methods; first, all proteins were cross-referenced for common contaminants, in which case they were removed from final analysis, and any polypeptides with less than two unique peptides identified were removed from final analysis. Low value imputation was used to substitute for missing values. Proteins identified in both biological replicates with ratio of recovery in +Ab greater than −Ab samples by a value of 5 or higher are reported in Supplementary Data 1.

### Reporting summary
Further information on research design is available in the Nature Portfolio Reporting Summary linked to this article.

## Data availability
The data that support this study are available from the corresponding author upon request. All raw sequencing files as well as processed data (bigwig files) have been deposited to GEO under record GSE218590. Mass spectrometry data have been deposited to the PRIDE database under project accession PXD045033. Source data are provided with this paper.

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

## Acknowledgements

We thank members of the Paull laboratory for helpful discussion and acknowledge NIH grants R01GM138548, P01CA092584, R35 GM122569, U01 DK127432, and NSF grant EFMA 193303 for support. A.M.-G. is a Howard Hughes Medical Institute (HHMI) awardee of the Life Sciences Research Foundation. T.H. is an Investigator of the HHMI.

## Author contributions

R.D. contributed to conceptualization, methodology, validation, formal analysis, investigation, visualization, and review and editing of the manuscript. A.M.-G. contributed to methodology, validation, formal analysis, investigation, and review and editing of the manuscript. H.B. contributed to methodology, validation, formal analysis, and investigation. P.W. contributed to methodology, validation, formal analysis, and investigation. T.H. contributed to methodology, resources, supervision, and review and editing of the manuscript. T.P. contributed to conceptualization, methodology, validation, formal analysis, investigation, resources, data curation, writing and editing of the manuscript, visualization, supervision, project administration, and funding acquisition.

## Competing interests

The authors declare no competing interests.
