## [Peer Review File · Nature Communications]

Genome-wide analysis of DNA-PK-bound MRN cleavage products supports a sequential model of DSB repair pathway choiceReviewers' comments:

Reviewer #1 (Remarks to the Author):

Deshpande et al carried out a series of ChIP-based experiments to assess the dynamics of MRE11-RAD50-NBS1 (MRN) recruitment and DNA end resection initiation with respect to the binding of double-strand break ends by DNA-PK. Their GLASS-ChIP technique allows the observation of DNA fragments bound by DNA-PK at DNA double strand breaks through the endonuclease activity of MRN. The authors used light-induced Cas9 to make breaks with high temporal resolution, which revealed MRN and DNA-PK are both present early (~15 min) after Cas9 activation, but release of the MRN-cut, DNA-PK-bound fragment occurs later (~60 min). These results suggest that there is less competition between DNA end joining and recombination machinery than is typically described in the field. These findings add valuable details to previously published work of these authors suggesting co-operative action of DNA double strand break repair proteins in end joining and recombination (Deshpande et al, 2020, *Sci. Adv.*).

Specific Comments

1. On line 109, it suggests that the DNA fragments observed at AsiSI sites are 150-200 bp on each side based on the inset from Fig. 1C, but the inset suggests 400 bp on each side of the DNA-PK peak. Please clarify which region of the peak is highlighted in the inset.
2. Does DNA-PK remain bound to DNA ends for extended periods of time when the endonuclease inhibitor of MRN is used (pellet-ChIP)?
3. How do the dynamics between MRN and DNA-PK change when the DSB repair pathway is impaired? For example, does DNA-PK enrichment still occur in 53BP1-deficient cells? What about BRCA1-deficient cells? CtIP?

Minor Comments

1. The green bars in Fig. 1C presumably indicate AsiSI sites, but it's not mentioned in the legend. Please indicate clearly.
2. There are multiple examples of points made but not explained. For example, lines 132-136 point out the observation that fragment length varies between conditions, but the point is not elaborated on. It would be helpful to fit these points in the context they are brought up in, not just the discussion.
3. Line 192, "ChIP" is written as "CHIP."

Reviewer #2 (Remarks to the Author):

In this manuscript the authors have examined the DNA-PK-bound DNA fragments that are processed by MRE11-RAD50-NBS1 (MRN) complex. The authors have used a previously described technique called GLASS-ChIP to identify and sequence MRN processed DSB fragments that can be coimmunoprecipitated using DNA PK phospho-S2056 antibody. Double strand breaks are generated using the AsiSI-ER DivA system, that allows generation of DSB by AsiSI restriction enzyme in the presence of tamoxifen. Examination of the fragments have revealed the width of the fragments to be around 150-200bp. It is interesting but unclear how inhibition of DNA-PK activity enhances the yield of the products but in the absence of the inhibitor, the yield is low, but the fragments are spread further away from the cut site (200bp vs. 1kb). Effect of cell cycle stage on GLASS-ChIP fragment recovery has also been analyzed. Use of light induced Cas9 has revealed that both MRE11 and DNA-PK bind to the DNA ends before the DNA -PK bound fragments are released.

All the experiments are performed well, and proper controls are used. Results are clear and interpreted correctly. However, a major concern is that none of the key findings are novel. The present work is focused on sequencing and characterizing the DNA PK bound DNA fragments that are cleaved by the MRN complex. The key conclusions from this study were reported in a previous study (Deshpande et al., in *Science Advances* 2020). The significance on DNA-PK binding to the repair of DSB by HR that is initiated by the processing of the broken ends by the MRN complex has

been described before. Importance of the endonuclease and not the exonuclease activity of MRE11 in the processing of the DSB has also been reported previously. Effect of DNA-PK inhibition on generation of DNA fragments has also been examined and reported previously. Lack of significant novelty is the only major concern.

A few minor concerns/suggestions:

1. Figure 4: Given the essential role of CtIP phosphorylation on MRN-mediated cleavage, CtIP-ChIP will be a good control to see if its recruitment at the site of AsiSI cut site coincides with MRE11.
2. Figure 5C: What was the rationale for selected the 4 genes (sites) to examine the effect of cell cycle on GLASS-ChIP and Pellet ChIP fragments. Why were these genes/sites not examined for resection studies in Figure S4 instead of KYAT3?
3. Figure 5C: Given the increase in HR in S and G2 phase, why isn't there any increase in GLASS-ChIP fragments in G2. There is a clear increase in the Pellet ChIP fragments of some genes in G1 and G2 and not of some of the other genes. This is predicted to reflect an increase in transcription of the genes with high Pelle ChIP. Authors should examine the transcription level of these genes at different cell cycle stages to make it more convincing.
4. Figure 8 is missing. Figure legend is included, and figure is cited but the figure is missing.
5. Figure 6D is not cited in the text

Reviewer #3 (Remarks to the Author):

The results presented in this manuscript extend upon the in vitro finding of a potential role for DNA-PK in stimulating MRN previously published by the authors. The manuscript provides a step forward in our understanding of the coordination between the NHEJ and HR repair pathways, for which, while there is some previous evidence, the mechanism is still not clearly understood. The first part of the paper (using the ER-fused AsiSI) recapitulates some aspects of previous results from the same authors. By inducing site-specific DSBs in the genome using the ER-fused AsiSI or light-activated Cas9, and using a modified GLASS-ChIP method, they measure the release of DNA-PK bound DNA fragments from DSB sites and refer to these as intermediates of end-processing. DNA-PK acts as a protein block at DNA ends, stimulating Mre11-mediated end-resection. Furthermore, this resection is dependent on the endonuclease activity of Mre11. The authors propose a sequential rather than competitive model of Mre11 and DNA-PK activity at breaks wherein both proteins bind the same end and DNA-PK stimulates Mre11 end-resection activity. The systems and approaches are elegant and the work is rigorous.

Main points to be addressed:

- 1) For most experiments, the authors inhibit DNA-PKs because the yields of released intermediates were higher. In the absence of the inhibitor, end-resection was heterogenous and more long-range, even though the yield is low (Fig 3B). It is not clear whether DNA-PKi is somehow modifying the behavior of repair factors on the DNA and only enriching for short-range resection. Furthermore, the short-range resection is partially dependent on Mre11 (Fig 2B). Is it possible that inhibiting DNA-PK and not binding of DNA-PK stimulates Mre11-mediated short-range resection? Showing that the long-range resection (without DNA-PKi) is also dependent on Mre11 endonuclease activity would strengthen the claim that DNA-PK stimulates Mre11 activity on DSBs.
- 2) In Fig 2B, resection is partially dependent on Mre11 endonuclease activity. Could the authors provide additional evidence or comment on the residual resection observed. Are there other factors that could be involved in DNA-PK dependent end-resection at DSBs? Furthermore, the authors could examine whether Mre11-dependent resection is cell cycle dependent.
- 3) Throughout the study, we observe a difference in resection patterns in the presence of DNA-PKi, specifically elimination of long-range resection, which strengthens the authors' claims that end-

resection is stimulated by DNA-PK. Could the authors show a complementary approach by knocking down Ku70 to show that Mre11 binding at the DSBs is dependent on DNA-PK end-binding.

4) In Fig 5A, authors show a correlation with HR factor RAD51 and NHEJ factors XRCC4 and Lig4 at DSBs. The authors should discuss these results further as this would suggest that cells rely on both NHEJ and HR for downstream repair.

5) The data in Fig 7A show very low levels for DNA-PKcs pellet-ChIP at 15 minutes but very strong at 60 minutes. Why is it so low at 15 minutes? If the "DNA-PKcs-bound product is released subsequent to this co-localization" why is the signal so high at 60 minutes? Are there sites where DNA-PKcs persists? Could the authors add a 60 minute plot to Fig 7C to visualize how the binding of DNA-PK and Mre11 change as more DNA-PK bound product is released?

6) While previous reports have looked at DNA-PK and Mre11 recruitment after laser irradiation (J Cell Biol. 2005 Aug 1;170(3):341-7. doi: 10.1083/jcb.200411083. PMID: 16061690; PMCID: PMC2171485) as well as more recently, recruitment of RPA in the presence or absence DNA-PK (Elife. 2022 May 16;11:e74700. doi: 10.7554/eLife.74700. PMID: 35575473; PMCID: PMC9122494.), the Cas9 system generates DSBs at a more physiological level and it could add to the results significantly to explore recruitment of downstream repair factors (HR vs NHEJ) at these DSBs in the presence or absence of NHEJ. As the title and the text strongly suggest a 'sequential' model of DSB repair, it would be important to determine if DNA-PK and Mre11 indeed bind the same sites and if DNA-PK is required for Mre11 binding and downstream recruitment of repair factors (RPA/Rad51).

Minor comments:

- 1) It would be interesting to add a panel of GC+4-OHT+DNA-PKi in Fig 4C, to highlight the effect of the inhibitor on GLASS-ChIP.
- 2) Fig 7A: Y-axis label (Cas9 sites) is on top of the figure and needs to be moved to the left.
- 3) Refs 22, 51 and 59 are incomplete.

Reviewers' comments (blue):

Reviewer #1 (Remarks to the Author):

Deshpande et al carried out a series of ChIP-based experiments to assess the dynamics of MRE11-RAD50-NBS1 (MRN) recruitment and DNA end resection initiation with respect to the binding of double-strand break ends by DNA-PK. Their GLASS-ChIP technique allows the observation of DNA fragments bound by DNA-PK at DNA double strand breaks through the endonuclease activity of MRN. The authors used light-induced Cas9 to make breaks with high temporal resolution, which revealed MRN and DNA-PK are both present early (~15 min) after Cas9 activation, but release of the MRN-cut, DNA-PK-bound fragment occurs later (~60 min). These results suggest that there is less competition between DNA end joining and recombination machinery than is typically described in the field. These findings add valuable details to previously published work of these authors suggesting co-operative action of DNA double strand break repair proteins in end joining and recombination (Deshpande et al, 2020, Sci. Adv.).

Thank you for the positive comments, particularly the fact that this study adds valuable details to our previously published work.

Specific Comments

1. On line 109, it suggests that the DNA fragments observed at AsiSI sites are 150-200 bp on each side based on the inset from Fig. 1C, but the inset suggests 400 bp on each side of the DNA-PK peak. Please clarify which region of the peak is highlighted in the inset.

The text refers to the average size of all the peaks found at all cleaved sites, which gives a value of 150 to 200 bp at 50% of the average height of the peak (Fig. 1B). The inset in Fig. 1C shows one specific peak which is a little wider than that average value. This will be made more clear in the text.

2. Does DNA-PK remain bound to DNA ends for extended periods of time when the endonuclease inhibitor of MRN is used (pellet-ChIP)?

Use of the endonuclease inhibitor for MRN at the same time as AsiSI induction results in a high level of toxicity, such that we need to limit the time of the assay to 1 hr. We would predict that the DNA-PK would remain bound for a significant time but the toxicity issue prevents us from determining this experimentally.

3. How do the dynamics between MRN and DNA-PK change when the DSB repair pathway is impaired? For example, does DNA-PK enrichment still occur in 53BP1-deficient cells? What about BRCA1-deficient cells? CtIP?

This is a great question about 53BP1-deficient cells and we are planning experiments now that address this, but we anticipate that this will be a separate project since 53BP1 and Brca1 and their functional interactions will involve extensive additional experiments.

Minor Comments

1. The green bars in Fig. 1C presumably indicate AsiSI sites, but it's not mentioned in the legend. Please indicate clearly.

2. There are multiple examples of points made but not explained. For example, lines 132-136 point out the observation that fragment length varies between conditions, but the point is not

elaborated on. It would be helpful to fit these points in the context they are brought up in, not just the discussion.

Additional discussion of the effect of DNA-PKCs inhibitor on DNA-PK mobility and occupancy has been added to the main text at this point in the Results section.

3. Line 192, “ChIP” is written as “CHIP.”

This has been corrected.

Reviewer #2 (Remarks to the Author):

In this manuscript the authors have examined the DNA-PK-bound DNA fragments that are processed by MRE11-RAD50-NBS1 (MRN) complex. The authors have used a previously described technique called GLASS-ChIP to identify and sequence MRN processed DSB fragments that can be coimmunoprecipitated using DNA PK phospho-S2056 antibody. Double strand breaks are generated using the AsiSI-ER DivA system, that allows generation of DSB by AsiSI restriction enzyme in the presence of tamoxifen. Examination of the fragments have revealed the width of the fragments to be around 150-200bp. It is interesting but unclear how inhibition of DNA-PK activity enhances the yield of the products but in the absence of the inhibitor, the yield is low, but the fragments are spread further away from the cut site (200bp vs. 1kb). Effect of cell cycle stage on GLASS-ChIP fragment recovery has also been analyzed. Use of light induced Cas9 has revealed that both MRE11 and DNA-PK bind to the DNA ends before the DNA -PK bound fragments are released.

All the experiments are performed well, and proper controls are used. Results are clear and interpreted correctly. However, a major concern is that none of the key findings are novel. The present work is focused on sequencing and characterizing the DNA PK bound DNA fragments that are cleaved by the MRN complex. The key conclusions from this study were reported in a previous study (Deshpande et al., in Science Advances 2020). The significance on DNA-PK binding to the repair of DSB by HR that is initiated by the processing of the broken ends by the MRN complex has been described before. Importance of the endonuclease and not the exonuclease activity of MRE11 in the processing of the DSB has also been reported previously. Effect of DNA-PK inhibition on generation of DNA fragments has also been examined and reported previously. Lack of significant novelty is the only major concern.

We appreciate the attention to detail here and the comments about the rigor of the experiments. As the reviewer points out, we analyze not only that the fragments are produced, but how they change in location with inhibition of DNA-PK catalytic activity and with cell cycle phase. In addition, we use the light-activated Cas9 system to show how the kinetics of bound and released Cas9 change in cells relative to that of Mre11. Our original paper had no sequencing data at all, but our current study shows genome-wide details for all experimental conditions, both the released and bound DNA-PK, as well as ChIP from the Cas9 light-activated system. In addition to this original data, we also now include CtIP ChIP-seq results that clearly show a correlation between the occupancy of this HR factor with NHEJ repair enzymes. These data show the dynamics of DNA-PK, Mre11, and CtIP at double-strand breaks with nucleotide resolution, and the results support a new sequential model of DSB repair.

A few minor concerns/suggestions:

1. Figure 4: Given the essential role of CtIP phosphorylation on MRN-mediated cleavage, CtIP-ChIP will be a good control to see if its recruitment at the site of AsiSI cut site coincides with

MRE11.

To address this we tested different CtIP antibodies and were able to find one that is functional in CHIP experiments. We performed CHIP-seq for CtIP in the absence of DNA-PK inhibition and found that this factor localizes very specifically to AsiSI sites with 4-OHT addition (Reviewer Fig. 1 and new Fig. 5 in the manuscript). Comparisons of this signal with DNA-PK pellet CHIP show that CtIP does not associate generally with promoter regions as observed with DNA-PKcs but appears specifically at DSB sites with 4-OHT addition (Reviewer Fig. 1B). Released fragments were analyzed for CtIP association as well ("CtIP GC") but no signal was observed, thus CtIP does not appear to be associated with the DNA-PK-bound ends that are released from the chromatin.

CtIP CHIP signal at AsiSI DSB sites extends even further than DNA-PKcs, as far as several kb distal from the DSB, although the majority of the mapped reads are within 2.5 kb of the break site (Reviewer Fig. 1A, B below). This accumulated signal suggests an involvement of the protein not only in the initiating events close to the break that promote DNA end resection but also potentially in long-range resection activities.

Quantitative analysis of the accumulation of CtIP CHIP signal at AsiSI cut sites shows a strong correlation with patterns of DNA-PKcs GLASS-CHIP ($\rho=0.69$), much higher than the correlation with chromatin-associated DNA-PKcs ($\rho=0.24$) (Reviewer Fig. 1D, E), consistent with our previous observations that CtIP collaborates with MRN and is essential for the endonuclease-mediated release of DNA-PK-bound ends¹. The highest correlation with CtIP CHIP is observed with previously published Xrcc4 and Rad51 CHIP data² (Reviewer Fig. 1F, G) with correlation coefficients 0.81 and 0.77, respectively.

Reviewer Figure 1. CtIP ChIP patterns show extended binding around DSB site. (A) Recovery of CtIP pellet-ChIP products at the top 300 AsiSI cutting sites is shown, in the absence or presence of 4-OHT (4 hrs) as indicated. (B) Genome browser views of 2 AsiSI sites showing CtIP ChIP signal (both released fragments "CtIP GC" and pellet CtIP ChIP "CtIP PC") as well as DNA-PKcs pellet-ChIP signal ("DNA-PK PC") with 4-OHT addition. Insets show higher resolution views. Green marks indicate locations of AsiSI sites. (C) High resolution view of one CtIP ChIP peak. (D) to (G) Comparisons of accumulated ChIP signal at the top 300 AsiSI cut sites with CtIP ChIP, DNA-PK pellet-ChIP, DNA-PK GLASS-ChIP, Xrcc4 ChIP, and Rad51 ChIP with Pearson correlation coefficients as indicated. (G) includes designations of HR-prone sites (orange) and NHEJ-prone sites (black)². Xrcc4 and Rad51 data from previously published datasets².

Interestingly, we observed substantial CtIP signal at the center of the AsiSI cut site, extending across the 2 nt gap that is formed at fragments containing the AsiSI-generated 3' overhang (Reviewer Fig. 1C), in contrast to the absence of CHIP signal observed at DNA-PK-bound fragments (Fig. 1 and 4, main manuscript). To examine this quantitatively, we performed a statistical comparison of DNA-PK and CtIP mapped reads at AsiSI sites in comparison to previously reported BLESS signal³ which labels DSBs⁴. This analysis shows that the CtIP span ratio (proportion of mapped reads extending across the center 2 nt) is significantly higher than with DNA-PK CHIP in the absence of inhibitor and is comparable to that of LIG4 (Reviewer Fig. 2 and Fig. S3, main manuscript). The binding of DSB factors to sequences spanning the AsiSI site likely indicate situations where the factors remain bound for some time after the site has been repaired. This might be expected for enzymes involved in the final step of repair such as LIG4 in NHEJ, but is an unexpected finding for CtIP which is thought to function primarily in the initiating stages of end processing.

Reviewer Figure 2. Analysis of ChIP-seq reads spanning the center of AsiSI and Cas9 cut sites. (A) Diagram of AsiSI recognition site and cut locations with examples of reads that do not span the cut site (grey) and reads that do span the cut site (pink). DNA fragments containing the 2 nt 3' overhang generated by AsiSI would be removed by treatment with the end repair enzyme mix (NEB) during library preparation, thus generating a gap in processed reads for proteins bound to an unrepaired, AsiSI-cut end. (B) Diagram of Cas9 GG multi-site target with reads shown as in (A). Cas9 generates blunt ends (black) but can also generate a staggered cut as shown (grey). (C) Read "span ratios" at the 50 top AsiSI cut sites for each respective ChIP experiment. The 50 top AsiSI cut sites analyzed here are the 50 cut sites that had the highest total read count across all ChIP experiments. Span ratio is defined as the ratio of the number of reads that span an AsiSI cut site divided by the total number of reads that fall within 5nt upstream or downstream of a cut site. Pairwise Kolmogorov Smirnov statistics were determined with 1,000 bootstraps for each pairwise comparison listed. (D) Read "span ratios" for the top 50 Cas9 target sites analyzed for BLISS, DNA-PK pellet-ChIP, Mre11 pellet-ChIP with DNA-PKi, and Mre11 pellet-ChIP (no DNA-PKi) datasets after light exposure as indicated. NS=non-significant; **= p value <0.001 , ***= p value <0.0001 . BLESS and LIG4 ChIP data³ and BLISS and Mre11 ChIP (no DNA-PKi) data^{5,6} are from previously published datasets.

2. Figure 5C: What was the rationale for selected the 4 genes (sites) to examine the effect of cell cycle on GLASS-ChIP and Pellet ChIP fragments. Why were these genes/sites not examined for resection studies in Figure S4 instead of KYAT3?

The sites we use for resection are the sites originally used in the development of the resection method and depend on restriction enzyme cut sites at informative locations relative to the AsiSI site. To enable comparison between the resection data and ChIP-qPCR results we added a graph showing ChIP-qPCR at the KYAT3 locus to this figure (now Fig. S4). The recovery of fragments is lower than at other AsiSI sites but the overall pattern is similar.

3. Figure 5C: Given the increase in HR in S and G₂ phase, why isn't there any increase in GLASS-ChIP fragments in G₂. There is a clear increase in the Pellet ChIP fragments of some genes in G₁ and G₂ and not of some of the other genes. This is predicted to reflect an increase in transcription of the genes with high Pellet ChIP. Authors should examine the transcription level of these genes at different cell cycle stages to make it more convincing.

We do not know why there is not a significant increase in recovery of released DNA-PK-bound fragments in G₂ phase. It could be that there are degradation pathways that target the released fragments, but we do not have any information about this yet. The histone and ribosome biogenesis-related genes H2AC21 and SNORD3A that we monitored in this figure (now Fig. S4) increase in transcriptional activity in S and G₂ phases of the cell cycle^{7,8}, perhaps related to the increase in pellet ChIP recovery, although our experiments do not show that definitively since we did not address the role of transcription in modulating the efficiency of DNA-PK-associated cutting by MRN.

4. Figure 8 is missing. Figure legend is included, and figure is cited but the figure is missing.

This has been corrected.

5. Figure 6D is not cited in the text

This has been corrected.

Reviewer #3 (Remarks to the Author):

The results presented in this manuscript extend upon the in vitro finding of a potential role for DNA-PK in stimulating MRN previously published by the authors. The manuscript provides a step forward in our understanding of the coordination between the NHEJ and HR repair pathways, for which, while there is some previous evidence, the mechanism is still not clearly understood. The first part of the paper (using the ER-fused AsiSI) recapitulates some aspects of previous results from the same authors. By inducing site-specific DSBs in the genome using the ER-fused AsiSI or light-activated Cas9, and using a modified GLASS-ChIP method, they measure the release of DNA-PK bound DNA fragments from DSB sites and refer to these as intermediates of end-processing. DNA-PK acts as a protein block at DNA ends, stimulating Mre11-mediated end-resection. Furthermore, this resection is dependent on the endonuclease activity of Mre11. The authors propose a sequential rather than competitive model of Mre11 and DNA-PK activity at breaks wherein both proteins bind the same end and DNA-PK stimulates Mre11 end-resection activity. The systems and approaches are elegant and the work is rigorous.

Thank you for the positive comments, particularly the statements noting how these results extend our initial in vitro findings.

Main points to be addressed:

1) For most experiments, the authors inhibit DNA-PKcs because the yields of released intermediates were higher. In the absence of the inhibitor, end-resection was heterogeneous and more long-range, even though the yield is low (Fig 3B). It is not clear whether DNA-PKi is somehow modifying the behavior of repair factors on the DNA and only enriching for short-range resection. Furthermore, the short-range resection is partially dependent on Mre11 (Fig 2B). Is it possible that inhibiting DNA-PK and not binding of DNA-PK stimulates Mre11-mediated short-range resection? Showing that the long-range resection (without DNA-PKi) is also dependent on Mre11 endonuclease activity would strengthen the claim that DNA-PK stimulates Mre11 activity on DSBs.

Experiments performed in the Jeggo and Tainer labs in their characterization of the Mre11 endo- and exonuclease inhibitors showed that blocking Mre11 catalytic activity impairs long-range end resection⁹. These experiments were performed with DNA-PKcs activity at normal levels. From our in vitro experiments we also know that the presence of DNA-PKcs protein on DNA ends (with Ku) stimulates MRN endonucleolytic cutting, in the absence of DNA-PK inhibitor¹. We also observed this stimulatory activity in single-molecule DNA curtains experiments in the absence of DNA-PK inhibitor, with MRN nuclease activity essential for the removal of DNA-PK from DNA ends. Taken together, these results suggest that DNA-PK and its presence on DNA ends promotes MRN activity as well as long-range resection.

2) In Fig 2B, resection is partially dependent on Mre11 endonuclease activity. Could the authors provide additional evidence or comment on the residual resection observed. Are there other factors that could be involved in DNA-PK dependent end-resection at DSBs? Furthermore, the authors could examine whether Mre11-dependent resection is cell cycle dependent.

The residual resection observed in the presence of the PFM01 inhibitor is likely due to the technical challenges with the toxicity of the inhibitor and the need to use high levels of it to achieve inhibition. We have not performed a detailed analysis of Mre11 dependence in different cell cycle phases beyond the qPCR and resection analysis shown in the manuscript. We are working toward a degron system which would allow us to monitor these outcomes more efficiently in the absence of the inhibitors.

3) Throughout the study, we observe a difference in resection patterns in the presence of DNA-PKi, specifically elimination of long-range resection, which strengthens the authors' claims that end-resection is stimulated by DNA-PK. Could the authors show a complementary approach by knocking down Ku70 to show that Mre11 binding at the DSBs is dependent on DNA-PK end-binding.

To address this question we tried depleting Ku, but the micromolar levels of the protein in cells prevents us from achieving a functionally useful depletion. Also the fact that complete removal of Ku in human cells is cell-lethal¹⁰ suggests that if we were able to deplete Ku entirely, the cells would not be viable.

4) In Fig 5A, authors show a correlation with HR factor RAD51 and NHEJ factors XRCC4 and Lig4 at DSBs. The authors should discuss these results further as this would suggest that cells rely on both NHEJ and HR for downstream repair.

Yes, we do find that both HR and NHEJ factors appear at a subset of locations genomewide and that they seem to be located in the same group of regions rather than separated into sites that prefer one or the other pathway. We have added additional analysis of this in the new figure on CtIP (Fig. 5) and also in Fig. 7 where we analyze Mre11 occupancy relative to DNA-PK. We find that CtIP is strongly correlated with both Rad51 and Xrcc4 ($\rho= 0.77$ and 0.81 , respectively) in addition to DNA-PK. In the Cas9 experiments, we found that Mre11 occupancy in the genome is also strongly correlated with DNA-PKcs at the 60 min. time point post light exposure ($\rho=0.87$). These overall results suggest a strong correlation between HR and NHEJ factors at DSB sites.

5) The data in Fig 7A show very low levels for DNA-PKcs pellet-ChIP at 15 minutes but very strong at 60 minutes. Why is it so low at 15 minutes? If the “DNA-PKcs-bound product is released subsequent to this co-localization” why is the signal so high at 60 minutes? Are there sites where DNA-PKcs persists? Could the authors add a 60 minute plot to Fig 7C to visualize how the binding of DNA-PK and Mre11 change as more DNA-PK bound product is released?

A 60 min. plot has been added to Fig. 7C, also shown below in Reviewer Figure 3. Note that the data has all been normalized for read depth in this version of the manuscript so the absolute values of the signals have changed since the previous version. The accumulation of phospho-DNA-PK in the chromatin pellet increases dramatically between the 15 and 60 min time points. To visualize all the data in the same plot, we have used a separate Y axis for the pellet ChIP at the 60 min time point. It is clear that the Mre11 ChIP pattern has the same relationship to the DNA-PK ChIP pattern at 60 min. as compared to the 15 min. time point (i.e. Mre11 binds distal to DNA-PK). As to why there is such high accumulation of DNA-PK at the 60 min. compared to 15 min. time point, we don't know but it is likely related to the process of Cas9 removal. Cas9 has been shown to remain bound to its target site after cleavage^{11,12} and the mechanisms of release are not fully understood¹³.

Reviewer Figure 3. Accumulation of Mre11 distal to DNA-PK at sites of light-activated Cas9 breaks. View of Mre11 ChIP and DNA-PKcs pellet ChIP (PC) signal at 15 min. and 60 min. after light exposure. Note separate Y axis for DNA-PK pellet ChIP at 60 min.

6) While previous reports have looked at DNA-PK and Mre11 recruitment after laser irradiation (J Cell Biol. 2005 Aug 1;170(3):341-7. doi: 10.1083/jcb.200411083. PMID: 16061690; PMCID: PMC2171485) as well as more recently, recruitment of RPA in the presence or absence DNA-PKi (Elife. 2022 May 16;11:e74700. doi: 10.7554/eLife.74700. PMID: 35575473; PMCID: PMC9122494.), the Cas9 system generates DSBs at a more physiological level and it could add to the results significantly to explore recruitment of downstream repair factors (HR vs NHEJ) at these DSBs in the presence or absence of NHEJ. As the title and the text strongly suggest a ‘sequential’ model of DSB repair, it would be important to determine if DNA-PK and Mre11 indeed bind the same sites and if DNA-PK is required for Mre11 binding and downstream recruitment of repair factors (RPA/Rad51).

Additional analysis of the levels of Mre11 and DNA-PK binding to DSB sites in the Cas9 light-activated system have been added to Fig. 7 and is also shown below in Reviewer Fig. 4. This shows that there is substantial correlation between Mre11 ChIP accumulation at the 60 min.

time point with both DNA-PKcs GLASS-ChIP and pellet ChIP signal ($\rho=0.81$ and $\rho=0.87$, respectively). The pellet ChIP signal for DNA-PK also correlates with Mre11 ChIP accumulation at the 15 min. time point ($\rho=0.81$).

Reviewer Fig. 4. Correlation between Mre11 and DNA-PKcs occupancy after light-activated Cas9 activation. (A) Comparison of accumulated Mre11 pellet ChIP signal at the 126 Cas9 target sites compared to DNA-PK GLASS-ChIP signal at 60 min. post light exposure with Pearson correlation coefficient as indicated. (B, C) Comparisons of accumulated Mre11 pellet ChIP signal at the 126 Cas9 target sites compared to DNA-PK pellet ChIP signal at 15 and 60 min. post light exposure, respectively, with Pearson correlation coefficients as indicated.

Minor comments:

- 1) It would be interesting to add a panel of GC+4-OHT+DNA-PKi in Fig 4C, to highlight the effect of the inhibitor on GLASS-ChIP.
- 2) Fig 7A: Y-axis label (Cas9 sites) is on top of the figure and needs to be moved to the left.
- 3) Refs 22, 51 and 59 are incomplete.

A GC +4OHT DNA-PKi track has been added to Fig. 4C.
The Fig. 7 Y axis label and the references have been fixed.

References cited

1. Deshpande, R. A. *et al.* DNA-dependent protein kinase promotes DNA end processing by MRN and CtIP. *Sci Adv* **6**, eaay0922 (2020).
2. Aymard, F. *et al.* Transcriptionally active chromatin recruits homologous recombination at DNA double-strand breaks. *Nat. Struct. Mol. Biol.* **21**, 366–374 (2014).
3. Clouaire, T. *et al.* Comprehensive Mapping of Histone Modifications at DNA Double-Strand Breaks Deciphers Repair Pathway Chromatin Signatures. *Molecular Cell* **72**, 250-262.e6 (2018).
4. Crosetto, N. *et al.* Nucleotide-resolution DNA double-strand break mapping by next-generation sequencing. *Nat Methods* **10**, 361–365 (2013).
5. Zou, R. S. *et al.* Massively parallel genomic perturbations with multi-target CRISPR interrogates Cas9 activity and DNA repair at endogenous sites. *Nat Cell Biol* **24**, 1433–1444 (2022).
6. Liu, Y. *et al.* Very fast CRISPR on demand. *Science* **368**, 1265–1269 (2020).
7. Armstrong, C. & Spencer, S. L. Replication-dependent histone biosynthesis is coupled to cell-cycle commitment. *Proc. Natl. Acad. Sci. U.S.A.* **118**, e2100178118 (2021).
8. Tsai, R. Y. L. & Pederson, T. Connecting the nucleolus to the cell cycle and human disease. *FASEB j.* **28**, 3290–3296 (2014).
9. Shibata, A. *et al.* DNA double-strand break repair pathway choice is directed by distinct MRE11 nuclease activities. *Mol Cell* **53**, 7–18 (2014).
10. Li, G., Nelsen, C. & Hendrickson, E. A. Ku86 is essential in human somatic cells. *Proc. Natl. Acad. Sci. U.S.A.* **99**, 832–837 (2002).

11. Sternberg, S. H., Redding, S., Jinek, M., Greene, E. C. & Doudna, J. A. DNA interrogation by the CRISPR RNA-guided endonuclease Cas9. *Nature* **507**, 62–67 (2014).
12. Richardson, C. D., Ray, G. J., DeWitt, M. A., Curie, G. L. & Corn, J. E. Enhancing homology-directed genome editing by catalytically active and inactive CRISPR-Cas9 using asymmetric donor DNA. *Nat Biotechnol* **34**, 339–344 (2016).
13. Feng, Y.-L., Wang, M. & Xie, A.-Y. Target residence of Cas9: challenges and opportunities in genome editing. *GENOME INSTAB. DIS.* **3**, 57–69 (2022).

REVIEWERS' COMMENTS

Reviewer #2 (Remarks to the Author):

Authors have satisfactorily addressed all my concerns.

Reviewer #3 (Remarks to the Author):

The authors have done a good job addressing reviewer concerns. No further issues.